# Suppression of angiopoietin-like 4 reprograms endothelial cell metabolism and inhibits angiogenesis

Balkrishna Chaube [1,2,3], Kathryn M. Citrin[1,2,3,4], Mahnaz Sahraei[1], Abhishek K. Singh[1,2], Diego Saenz de Urturi[1,2,3], Wen Ding[1], Richard W. Pierce [2,5], Raaisa Raaisa[6], Rebecca Cardone[6], Richard Kibbey[3,4,6], Carlos Fernández-Hernando [1,2,3,7] & Yajaira Suárez [1,2,3,7] ✉

Angiopoietin-like 4 (ANGPTL4) is known to regulate various cellular and systemic functions. However, its cell-specific role in endothelial cells (ECs) function and metabolic homeostasis remains to be elucidated. Here, using endothelial-specific *Angptl4* knock-out mice (*Angptl4^{iΔEC}*), and transcriptomics and metabolic flux analysis, we demonstrate that ANGPTL4 is required for maintaining EC metabolic function vital for vascular permeability and angiogenesis. Knockdown of ANGPTL4 in ECs promotes lipase-mediated lipoprotein lipolysis, which results in increased fatty acid (FA) uptake and oxidation. This is also paralleled by a decrease in proper glucose utilization for angiogenic activation of ECs. Mice with endothelial-specific deletion of *Angptl4* showed decreased pathological neovascularization with stable vessel structures characterized by increased pericyte coverage and reduced permeability. Together, our study denotes the role of endothelial-ANGPTL4 in regulating cellular metabolism and angiogenic functions of EC.

Angiopoietin-like 4 (ANGPTL4) is a family member of eight proteins from related gene products that are structurally similar to the angiopoietins[1]. ANGPTL4 participates in an array of biological functions, including the regulation of lipid and glucose metabolism, wound healing, hematopoietic stem cell expansion, and angiogenesis[2,3]. Accordingly, ANGPTL4 has been implicated in many pathological disorders, including cardiometabolic diseases such as diabetes and atherosclerosis[4,5]. ANGPTL4 was discovered simultaneously by three different screenings: a screen of novel PPAR gamma targets, a screen for novel fasting-induced factors from liver, and a PCR screen to identify novel angiopoietin-related proteins[1,6,7]. Since its discovery, an ample body of literature supports that ANGPTL4 governs fatty acid (FA) delivery to cells by inhibiting lipoprotein lipase (LPL) activity[3,8,9].

Overexpression of *Angptl4* in mice leads to hypertriglyceridemia, whereas deficiency lowered circulating lipids and reduced risk of coronary artery disease in humans[2,10,11]. In line with these findings, most of the lipid regulating effects can be attributed primarily to ANGPTL4 inhibition of LPL and hepatic lipase (HL) from adipose tissues and liver, respectively, where it is highly expressed[12–16]. ANGPTL4 is also expressed in the vascular endothelium[17] where it has been reported to regulate many a priori non-metabolic functions[2]. Indeed, given its similarity in structure with angiopoietins, ANGPTL4 attracted attention as a potential angiogenic and vascular permeability factor, especially given its control of a multitude of endothelial cell (EC) functions including vascular inflammation, nitric oxide production, angiogenesis, and vascular permeability[18–22].

[1]Department of Comparative Medicine, Yale University School of Medicine, New Haven, CT, USA. [2]Vascular Biology and Therapeutics Program, Yale University School of Medicine, New Haven, CT, USA. [3]Yale Center for Molecular and System Metabolism, Yale University School of Medicine, New Haven, CT, USA. [4]Department of Cellular & Molecular Physiology, Yale University, New Haven, CT, USA. [5]Department of Pediatrics, Yale University School of Medicine, New Haven, CT, USA. [6]Department of Internal Medicine, Yale University, New Haven, CT, USA. [7]Department of Pathology, Yale University School of Medicine, New Haven, CT, USA. ✉e-mail: yajaira.suarez@yale.edu

During angiogenesis, ECs lose their quiescence, decrease barrier function, proliferate, degrade extracellular matrix, change their adhesive properties, migrate, form tube-like structures and eventually mature into new blood vessels[23]. Angiogenesis is a complex process involving many molecular and cellular events that require temporal and spatial orchestration by a finely tuned balance between stimulatory and inhibitory signals whose transduction pathways lead specific programs of gene expression to assure an adequate angiogenic response which has recently been shown to be driven by a metabolic switch[24,25]. For instance, hypoxia promotes vessel growth by upregulating multiple pro-angiogenic pathways and reinforcing EC glycolytic metabolism. Interestingly, EC expression of ANGPTL is induced by hypoxia, and initially described to induce a strong pro-angiogenic response[1,17,26]. However, contradictory results have been reported regarding its contribution to angiogenesis and vascular integrity[18,19,21,27–32]. The apparent discrepancies seem to derive from the use of different experimental approaches (i.e., the use of in vitro systems with different forms of ANGPTL4), and the use of global knockout animal models to address the effect on the endothelium. In addition, most of these studies only considered the paracrine actions of ANGPTL4 on ECs and neglected the potential of a specific lipase-dependent autocrine role of endothelial ANGPTL4.

ECs play a critical role in the homeostasis of systemic lipid metabolism via facilitating the lipoprotein lipase (LPL) dependent hydrolysis of triglyceride (TG)-rich lipoprotein particles and the transport of FA to parenchymal tissues[33,34]. Following TG hydrolysis, FAs are taken up by ECs via different FA transporters[35–37]. Several factors regulate FA uptake and transport in ECs; however, little is known about how lipid uptake and metabolism are controlled in EC. Importantly, how alterations in lipid metabolism impact EC functions needs further investigation. ANGPTL4 plays a vital role in lipid metabolism as an inhibitor of LPL; however, whether it affects cellular bioenergetics and FA metabolism in ECs is unknown. Indeed, the role of EC-derived ANGPTL4 in regulating endothelial lipase (EL) activity remains to be explored in the context of regulating the angiogenic function of ECs. To address these questions, we used inducible endothelial-specific KO mice for Angptl4 (Angptl4[iΔEC]) and in vitro ANGPTL4 knockdown approaches. Using transcriptomics and steady-state metabolic flux analysis, we uncovered distinct metabolic phenotypes in ECs lacking ANGPTL4, characterized by elevated FA uptake and utilization favoring FA oxidation. Here, we show that ANGPTL4 is necessary for maintaining the metabolic homeostasis required for EC proliferation and migration. Conditional loss of Angptl4 in ECs in vivo diminishes pathological angiogenesis by decreasing EC proliferation which is associated with lipase-mediated alterations in FA metabolism. Overall, we conclude that endothelial-ANGPTL4 participates in the regulation of EC angiogenic functions and vascular stability in a cell-autonomous manner.

## Results

### ANGPTL4 KD promotes an anti-angiogenic program

To understand the role of ANGPTL4 in ECs, we performed a gene expression profile analysis on human umbilical vein ECs (HUVECs) upon ANGPTL4 knockdown (KD) using an siRNA approach (siANGPTL4). As expected, siANGPTL4 efficiently reduced mRNA (Fig. 1a) and protein levels of intracellular or secreted ANGPTL4 (Fig. 1b, left and right panels) when compared to non-silencing control (NS) conditions. Multidimensional scaling (MDS) plot showed that NS and siANGPTL4 conditions clustered into two distinct groups (Fig. 1c, left panel), indicating distinct gene expression patterns with >5000 differentially expressed (DE) genes of which 2625 were upregulated and 2586 downregulated (FDR < 0.05, Log2 Fold Change ≥1.5) (Fig. 1c, right panel). ANGPTL4 KD did not alter the mRNA levels of the ANGPTL family members expressed or not in human ECs (Supplementary Fig. 1a). Gene Ontology (GO) term enrichment and KEGG pathway analysis of DE genes identified gene signatures involved in lipid metabolism (e.g., lipid transport, cellular responses to lipids and FAs), as well as FoxO and Hippo signaling pathways, were upregulated, whereas signatures implicated in glycolysis and angiogenesis (e.g., cell adhesion, cell migration, and growth factor signaling) were downregulated (Fig. 1d). Representative heatmaps are depicted in Fig. 1e-g. Interestingly, genes involved in endothelial pro-angiogenic responses (e.g., KDR, FGFR1, ANG) were significantly downregulated upon depletion. In line with this, we also noted that ANGPTL4 KD in normoxic conditions significantly transcriptionally downregulated the hypoxia-inducible factor alpha (HIF-1α) signaling pathway and glycolysis (Fig. 1d), which are implicated in angiogenesis[38]. Conversely, gene signatures associated with FoxO, Hippo, and lipid/FA metabolic pathways -described features associated with quiescent ECs[39–42]- were upregulated in ECs upon siANGPTL4 (Fig. 1d). Subsequently, we evaluated whether basic angiogenic functions, namely proliferation, migration, and morphogenesis (cord formation) were altered. ANGPTL4 KD reduced cell proliferation, quantified by cell counting and EdU incorporation (Fig. 1h; Supplementary Fig. 1b). However, cell viability, assessed as CCK8 and Annexin V/PI staining (Supplementary Fig. 1c, d), and senescence, determined by β-Galactosidase staining (Supplementary Fig. 1e), were not affected. siANGPTL4 diminished both basal and vascular endothelial growth factor A (VEGFA)-induced EC migration (Fig. 1i). We also found decreased cell adhesion to different matrix proteins (Fig. 1j; Supplementary Fig. 1f). Reduced adhesion to fibronectin and gelatin was also accompanied by decreased surface expression of integrins, especially αvβ3 (Supplementary Fig. 1g), which is involved in pro-angiogenic signaling[43]. ANGPTL4 KD also significantly impaired cord formation under basal and growth factor-stimulated conditions (Fig. 1k). These effects on angiogenesis were in line with the significant downregulation of the mRNA (Fig. 1e) and protein levels of KDR and accompanied with a reduction in VEGFA-mediated activation/phosphorylation of extracellular signal-regulated protein kinase (ERK)1/2 (Supplementary Fig. 1h). The angiogenic effects were further confirmed by a three-dimensional assay where ECs proliferate, migrate and sprout through a fibrin matrix[44]. siANGPTL4 significantly decreased the number of branches and sprouts as well as their length when compared to NS control (Supplementary Fig. 1i). These data indicate that EC ANGPTL4 positively regulates angiogenic functions.

### Angptl4 deletion diminishes pathological angiogenesis

ANGPTL4 has been described to have both pro- or anti-angiogenic functions. To determine the impact of the inactivation of endothelial Angptl4 on angiogenesis, we generated an inducible EC-specific Angptl4 knockout mouse (Angptl4[iΔEC]). We first determined the role of ANGPTL4 in neonatal physiological retinal angiogenesis in Angptl4[iΔEC] or respective control Angptl4[flox/flox] mice (hereafter referred to as WT mice). While previous work showed decreased retinal angiogenesis when Angptl4 was deleted in a global manner[32], we did not find any gross effect on radial outgrowth or endothelial coverage at different time points of retinal development (Supplementary Fig. 2). Thus, we wondered whether endothelial ANGPTL4 could instead play a role in adult and pathological angiogenesis. Tamoxifen-mediated Cre activation significantly reduced Angptl4 mRNA in adult ECs (Supplementary Fig. 3a) without affecting the expression of other ANGPTL family members expressed or not in mouse ECs (Supplementary Fig. 3b). We used the VEGFA-containing Matrigel plug angiogenesis assay. Plugs from Angptl4[iΔEC] mice macroscopically appeared avascular when compared to those from WT mice and exhibited diminished hemoglobin content (Supplementary Fig. 3c and d respectively), indicating inadequate neovascularization when compared to the WT plugs. Microscopic analysis confirmed a clear reduction of EC-containing vessel structures, as determined by isolectin B4 staining (Supplementary Fig. 3e), further supporting an underdeveloped

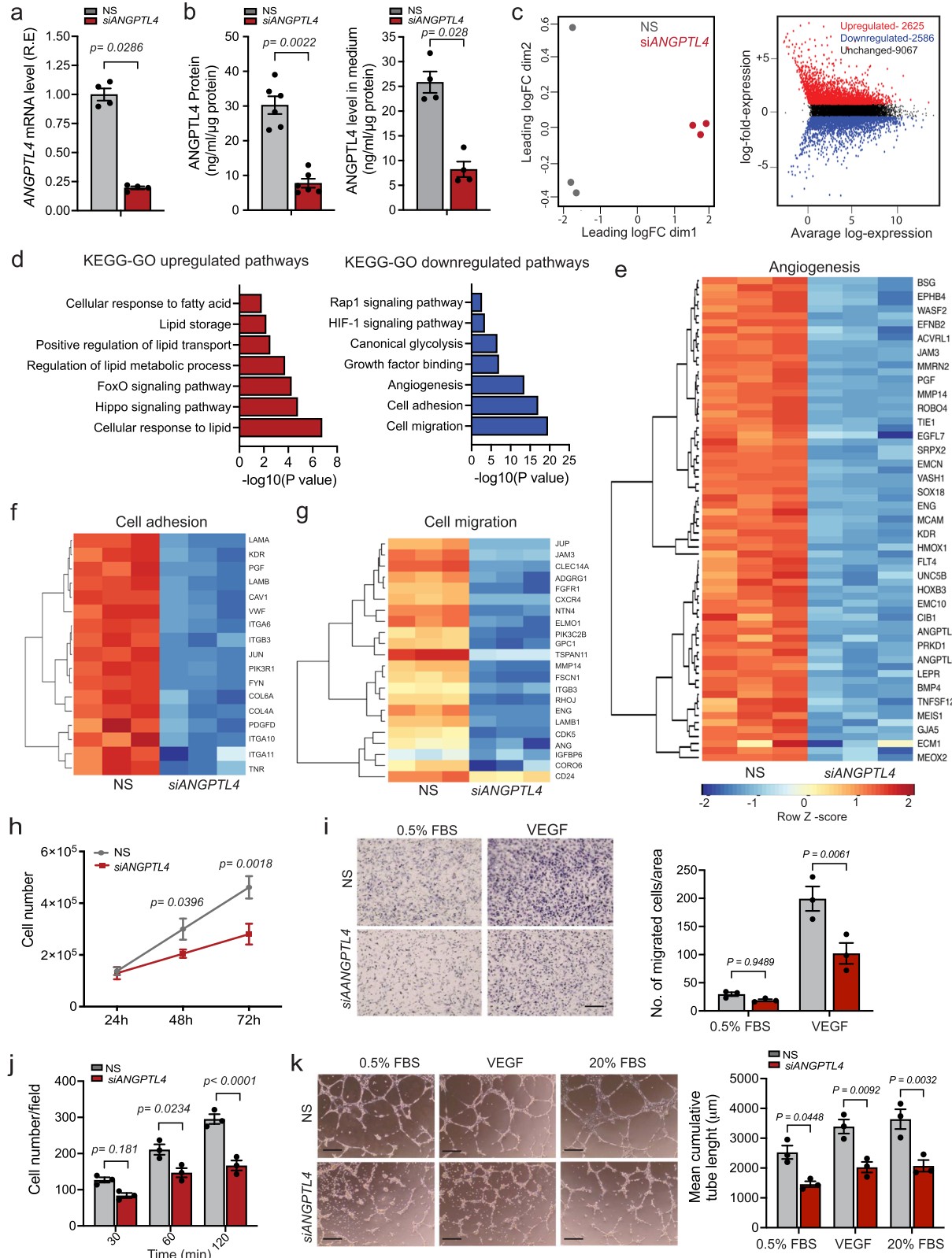

endothelial network in plugs from *Angptl4^iΔEC* mice. We then used Lewis lung carcinoma (LLC) and melanoma B16F10 isograft tumor models to study tumor-induced pathological angiogenesis and tumor progression[45,46]. We observed that *Angptl4^iΔEC* mice developed smaller tumors compared to WT (Fig. 2a; Supplementary Fig. 4a). This effect was accompanied by a reduction in CD31+ vessel-like structures (as seen in cross-sections; Fig. 2b; Supplementary Fig. 4b) and an overall

decrease in EdU+ ERG+ ECs as well as Ki-67+ CD31+ ECs (Fig. 2c, upper and lower panels, respectively), indicating diminished intratumor endothelial proliferation. Since increased vascular permeability is associated with tumor angiogenesis, tumor growth, and metastasis, and reports regarding the role of ANGPTL4 on vascular permeability are contradictory[2], we intravenously injected FITC-dextran of varying molecular weights or Evans Blue (EB) into tumor-bearing mice.

**Fig. 1 | Endothelial ANGPTL4 regulates EC Angiogenic Functions.** HUVECs were transfected for 6 h with siRNA against *ANGPTL4* or NS control, cells and/or media were harvested 60 h post-transfection unless otherwise indicated. **a** qRT-PCR analysis showing the expression of *ANGPTL4* (*n* = 4 independent experiments). **b** ELISA analysis of intracellular (left panel) and secreted (right panel) levels of ANGPTL4. (**a**, left panel, *n* = 6 and right panel *n* = 4 independent experiments). **c**–**g**, RNAseq analysis (*n* = 3 biological replicates). **c** MDS plot shows distinct clustering of the NS and si*ANGPTL4* samples (left panel). MD plot (right panel) showing the log-fold change and average abundance of each gene. Significantly altered genes (Fold change >1.5) of upregulated (2625 genes) and downregulated genes (2586) in *ANGPTL4* silenced HUVECs. **d** KEGG-GO pathway analysis of differentially upregulated (red bars) and downregulated (blue bars) genes in si*ANGPTL4* vs NS (significance for DGE was determined with Limma's decideTests). **e**–**g** Heatmaps of transcript levels of angiogenesis, cell adhesion, and cell migration pathway genes. Color key: red-high correlation; blue- low correlation. **h** Proliferation, alive cells

were counted at the indicated time points post transfection (*n* = 3 independent experiments performed in duplicate). **i** 60 h post-transfection, migration in basal (0.5%) FBS or in response to VEGF tested by Transwell cell Migration assay. Representative micrographs of migrated cells (left panel). Right panel, quantitation of number of cells migrated per area. **j**, Cell adhesion on gelatin-coated glass plates at the indicated time points, with cells harvested 60 h post-transfection. **k**, Matrigel cord formation assay. Cells were counted and seeded on a Growth Factor Reduced Matrigel in the presence of 0.5% FBS (basal; left panels), 0.5% FBS + VEGF (VEGF stimulated; middle panels) and 20% FBS (serum-stimulated; right panels). Cumulative sprout length of capillary-like structures measured by light microscopy after 8 h. Representative micrographs and quantification of cumulative tube lengths are shown. (**i**–**k**, *n* = 3 independent experiments. Scale bars, 75μm (**i**), 50μm (**k**). Data are represented as means ± SEM. Mann–Whitney U test in (**a** and **b**), and Two-way ANOVA with Tukey's multiple comparisons test in (**h**–**k**). Exact *p* values are shown for each comparison. Source data are provided as a Source data file.

Fluorescent imaging of both forms of FITC-dextran showed a lower diffusion in *Angptl4*[iΔEC] tumors (Fig. 2d and Supplementary Fig. 4c). Spectrophotometric measurement of extravasated EB showed significantly lower levels of EB in tumors from *Angptl4*[iΔEC] (Supplementary Fig. 4d). Accordingly, tumor vessels showed reduced intra-tumoral hemorrhage as indicated by the decreased extravascular Ter119[+] (red blood cell marker) area (Supplementary Fig. 4e). Mural/pericyte coverage was then assessed by immunofluorescence staining with alpha-smooth muscle actin (αSMA) and chondroitin sulfate proteoglycan (NG2[+]) antibodies. As shown in Fig. 2e and Supplementary Fig. 4f, CD31[+] tumor vessels from *Angptl4*[iΔEC] have significantly increased αSMA coverage with a modest alteration of pericyte coverage. Thus, increased barrier function of tumor vessels in *Angptl4*[iΔEC] mice was accompanied with enhanced vessel maturation and integrity, which are regulated by EC-pericyte/mural interactions. These results suggest that endothelial ANGPTL4 might facilitate tumor intravasation and metastasis, as described previously[31]. Since tumor cells also secrete ANGPTL4, and tumor-derived ANGPTL4 has been reported to exhibit both pro- and anti-tumor angiogenic properties[47], we aimed to more specifically investigate the role of endothelial ANGPTL4 in tumor-induced angiogenesis. Thus, we generated LLC cells deficient for *Angptl4*, referred as LLC[Angptl4-KD], using a short hairpin approach (Supplementary Fig. 5a) and, we implanted them into WT and *Angptl4*[iΔEC] mice. Importantly, the KD of *Angptl4* in these cells did not affect their proliferative response (Supplementary Fig. 5b). Briefly, when LLC[Angptl4-KD] cells were implanted into WT or *Angptl4*[iΔEC] mice, the latter, again, developed smaller tumors (Supplementary Fig. 5c), which was accompanied with a reduction in CD31[+] vessel-like structures (Supplementary Fig. 5d). These results were comparable to those obtained when LLC expressing *Angptl4* were implanted in WT or *Angptl4*[iΔEC] mice (Fig. 2a and b), and suggest a cell-autonomous role of endothelial ANGPTL4 in regulating tumor-induced angiogenesis, since in the absence of cancer cell-derived ANGPTL4, *Angptl4*[iΔEC] mice still developed smaller tumors compared to WT mice.

### *ANGPTL4* KD alters endothelial metabolism

ECs activated by angiogenic stimuli undergo significant metabolic rewiring, most notably an increase in glycolysis[24]. As described above, si*ANGPTL4* altered the metabolic transcriptional profile, particularly the downregulation of glycolysis and upregulation of lipid metabolism (Fig. 1d). Interestingly, despite normoxic culture conditions, *ANGPTL4* KD altered the hypoxia response gene signature (Fig. 1d). Hypoxia-induced transcriptional changes are associated with induced glycolysis and angiogenic responses[38]. Since *ANGPTL4* expression is induced by hypoxia[20,26,48], we wondered whether this induction helps to maintain a pro-angiogenic glycolytic phenotype, since its absence diminished angiogenesis and decreased the expression of glycolytic enzymes. Thus, we performed RNA-seq analysis in HUVECs upon silencing *ANGPTL4* treated with or without the hypoxia mimetic agent (CoCl₂) to

pathophysiologically induce ANGPTL4 levels. Analysis of the most variable genes in response to CoCl₂ stimulation in NS conditions confirmed upregulation of hypoxia-regulated genes (e.g., *LDHA*, *SLC2A1*, *EDN1*, *HK2*, among others), as well as *ANGPTL4* (Supplementary Fig. 6a). We also found that upon silencing *ANGPTL4*, a set of genes (1056 upregulated and 868 downregulated; FDR < 0.05, Log₂ Fold Change ≥1.5) were altered independently of CoCl₂ stimulation (Fig. 3a), suggesting that they were specifically regulated by ANGPTL4 in ECs. KEGG-GO pathway analysis of these genes revealed downregulation of gene signatures of glycolysis, pentose phosphate pathway, NAD metabolism, amino acid biosynthesis, and HIF1 signaling pathway (Fig. 3b), while FA metabolism and mitochondrial pathways were upregulated (Fig. 3b). These results suggest that ANGPTL4 regulates metabolic transcriptional programs in ECs linked to angiogenic responses. A more detailed analysis of the above pathways showed that silencing *ANGPTL4* prevented the hypoxia-mediated regulation of metabolic gene expression (Fig. 3c). Transcriptional changes of commonly regulated metabolic genes are summarized diagrammatically (Fig. 3d).

Next, we performed functional metabolic assays to complement the transcriptional changes seen upon *ANGPTL4* KD. First, we checked glucose uptake using a fluorescent derivative of glucose, 2NBDG, in basal or pro-angiogenic conditions (e.g., VEGFA treatment). As expected, in control conditions, VEGFA increased glucose uptake (Fig. 4a). Interestingly, *ANGPTL4* KD significantly decreased 2NBDG uptake in all conditions (Fig. 4a), which agrees with diminished *SLC2A1* (GLUT1) mRNA levels upon *ANGPTL4* KD (Supplementary Fig. 6b). Accordingly, *ANGPTL4* silencing markedly reduced glucose utilization and decreased lactate secretion (Fig. 4b and c), thus reflecting an overall reduction in glycolytic metabolism. We further verified these results by assessing extracellular proton flux through Seahorse analysis in *ANGPTL4* KD ECs followed by VEGFA stimulation or CoCl₂-induced pseudo-hypoxic-angiogenic stimulation. *ANGPTL4* depletion suppressed glycolysis, as indicated by reduced extracellular acidification rate (ECAR), under basal and stimulated conditions (Fig. 4d-f). Interestingly, upon *ANGPTL4* KD, the activity of hexokinase (HK2) and phosphofructokinase (PFK), rate-limiting glycolytic enzymes involved in EC function and angiogenesis[49,50], were considerably reduced (Fig. 4g), and in line with their diminished mRNA levels (Fig. 3d). Further validation at the mRNA and protein levels of key molecules regulating glycolysis in basal and stimulated conditions is shown in Supplementary Fig. 6b, c.

Based on these results, we presumed that *ANGPTL4* suppression in ECs might induce mitochondrial metabolism to sustain ATP generation. Thus, we checked oxygen consumption rate (OCR) and ATP generation from oxidative phosphorylation (OxPhos) using Seahorse analysis. We observed a significantly increased basal, ATP-linked, and maximal respiration capacity upon silencing *ANGPTL4* (Supplementary Fig. 6d). Of interest, mRNA levels of key tricarboxylic acid cycle (TCA)

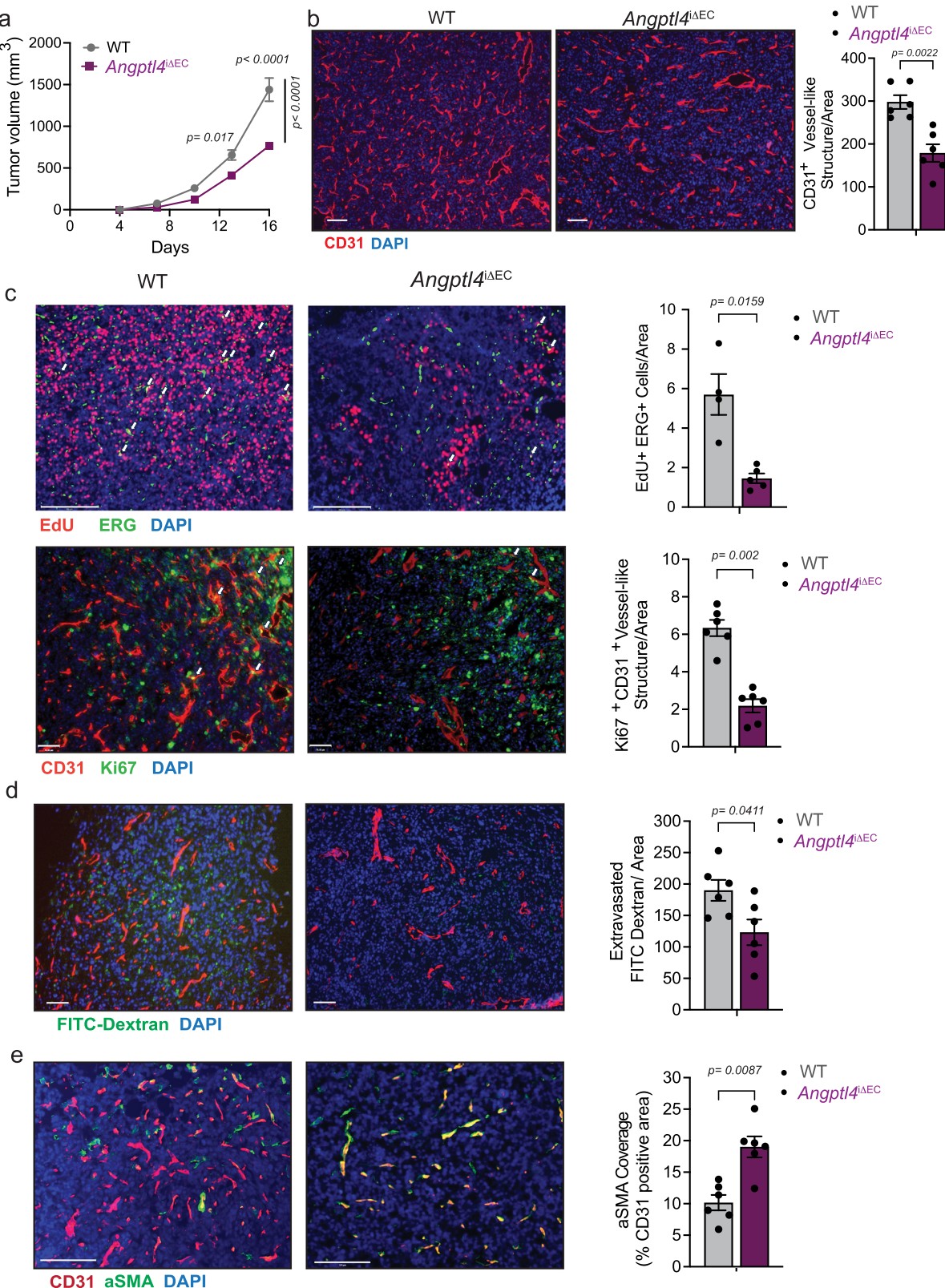

and OxPhos genes were slightly elevated upon *ANGPTL4* KD and after VEGFA or CoCl$_2$ stimulation and when compared to NS (Supplementary Fig. 6e). Overall, suppression of *ANGPTL4* abrogated the effect mediated by VEGFA or CoCl$_2$ on both glycolytic, TCA and OxPhos-related genes (Supplementary Fig. 6a-d). These results suggest that ANGPTL4 participates in maintaining EC metabolic function downstream of angiogenic stimuli, especially regarding glycolysis[24].

We performed Mass Isotopomer MultiOrdinate Spectral Analysis (MIMOSA) to analyze steady-state metabolites of glycolysis and TCA pathways and metabolic fluxes[51]. Using stable isotopes, this technique quantitatively measures the relative rates of sequential individual reactions in glycolytic and mitochondrial metabolism to identify differences in fuel preference (e.g., glycolysis vs. FA β-oxidation). As expected, we observed decreased levels of glycolytic intermediates

**Fig. 2 | EC-specific deletion of *Angptl4* reduces adult pathological angiogenesis in vivo.** **a**–**e** Tumor analysis of WT (*n* = 6) and *Angptl4*^iΔEC (*n* = 9) mice with s.c. injection of LLCs in the dorsal flank. **a** LLC isograft progression (tumor volume) in WT and *Angptl4*^iΔEC mice. **b** Left, representative micrographs of CD31 (red) and DAPI (blue) immunostaining. Right, quantification of CD31+ vessel-like structures. **c** Upper left, representative micrographs of immunofluorescence staining for EdU (red), and ERG (green), DAPI (blue). Proliferating (EdU and ERG double-positive) ECs are shown in yellow (arrow). Upper right, quantification of EdU and ERG double-positive cells (*n* = 4 for each genotype). Lower left, representative micrographs Ki67 (green) and CD31 (red) and DAPI immunostaining. Lower right, quantification of Ki67 and CD31 double positive cells shown in yellow (arrow)

(*n* = 6 for each genotype), **d** Left, representative images of FITC-Dextran (70 kDa) (green), CD31 (red) and DAPI immunostaining. Right, quantification of extravasated FITC-Dextran. **e** Left, representative micrographs of αSMA (green), CD31 (red) immunostaining. Right, quantification of αSMA covered CD31+ vessel-like structures. Right panels (**b**–**e**) quantification of at least 4 different images from each mouse, (*n* = 6 quantified per genotype out of 6 or 9 randomly selected, except for **c** (*n* = 4 or 5 per genotype were used). Scale bars, 70μm (**b**, **d**, and **c** lower panel), 125μm (**e**), 200μm (**c** upper panel). Data are represented as means ± SEM. Two-way ANOVA with Tukey's multiple comparisons test in (**a**) and Mann–Whitney U test in (**b**–**e**). Exact *p* values are shown for each comparison. Source data are provided as a Source data file.

(Fig. 4h). However, TCA cycle intermediates remained unaltered, except for elevated citrate (Fig. 4h) thus confirming the impaired glucose metabolism in *ANGPTL4* KD ECs observed with respirometry. Interestingly, we observed decreased $V_{PDH}/V_{CS}$ ratio (flux through pyruvate dehydrogenase *relative to* citrate synthase) upon *ANGPTL4* depletion (Fig. 4i), which together with the OCR data indicates an increase in relative and absolute rates of FA oxidation (FAO) as compared to control, suggesting preference for FA utilization. In line with these findings, immortalized mouse lung endothelial cells (MLECs) isolated from WT and *Angptl4*^iΔEC mice exhibited decreased mRNA levels of genes involved in glucose uptake and glycolysis including *Slc2a1* (GLUT1)*, Hk2, Pfkfb3, Ldha*, while the expression of genes involved in FAO, such as *Cpt1a* and *Acsl1*, was increased (Supplementary Fig. 6f).

To further assess whether the reduced glycolytic phenotype observed upon ANGPTL4 KD was, at least in part, dependent on its secretion from ECs, we incubated control or *ANGPTL4* KD ECs in conditioned media (CM) enriched (NS-Hypoxia CM) or depleted of ANGPTL4 (siANGPTL4-Hypoxia CM) (Supplementary Fig. 6g). Interestingly, when HUVECs with *ANGPTL4* KD were incubated in ANGPTL4-enriched CM, they exhibited increased glycolysis when compared to *ANGPTL4* KD HUVECs grown in CM lacking ANGPTL4 (Supplementary Fig. 6h). Similar effects were observed on proliferative responses (Supplementary Fig. 6i), strongly suggesting autocrine functions of endothelial ANGPTL4. These results, together with the increased rate of FAO and presumably increased FA utilization, led us to investigate lipase-mediated inhibitory actions of endothelial ANGPTL4.

### *ANGPTL4* KD increases EL activity, FA uptake and oxidation

ANGPTL4 negatively regulates LPL[3] as well as endothelial lipase (EL, gene name *LIPG*) which is expressed by both human and mouse ECs[52]. Our RNA-seq analysis showed an inverse correlation in the expression of *ANGPTL4* and *LIPG* in both basal and pseudo-hypoxic conditions (Supplementary Fig. 7a). Thus, we tested whether the effects of *ANGPTL4* on glycolysis could be mediated by its lipase inhibitory mechanism. Interestingly, upon *ANGPTL4* KD in the presence of lipoproteins naturally found in the media, we observed a significant elevation of lipase activity in the media from these cells (Fig. 5a). This effect was accompanied by increased FA uptake, FA translocase scavenger receptor (CD36) surface expression, and FAO (Fig. 5b-d). Based on these results, we hypothesized that the enhanced FA uptake observed upon *ANGPTL4* KD may be a causative factor for reduced glucose uptake and glycolysis: increased FA uptake/oxidation yields acetyl-CoA from which citrate is generated in the mitochondria. Citrate transport to the cytosol may then inhibit PFK and thus, glycolysis[53]. To address this hypothesis, we sought to validate whether *ANGPTL4* KD increases lipase activity in EC in the absence or presence of a well-described pharmacological inhibitor of EL activity, XEN455[54]. Importantly, mRNA levels of the two isoforms of EL (*LIPG1* and *LIPG2*) were both significantly increased upon *ANGPTL4* KD but unaltered by XEN455 (Supplementary Fig. 7b). In control conditions, where cellular lipase inhibitor (i.e., ANGPTL4) is present, XEN455 did not alter

endothelial triglyceride lipase (Fig. 5e, left) and phospholipase activity (Fig. 5e, right). However, *ANGPTL4* KD-mediated induction of lipase activity in ECs was significantly reduced in the presence of XEN455 (Fig. 5e). To evaluate triglyceride lipase activity during lipolysis in ECs, we used radiolabeled TAG-rich particles (chylomicrons) and triolein (Fig. 5f, left and right panels). Increased lipid uptake was observed in si*ANGPTL4* conditions, as compared to NS control (Fig. 5f). As above, XEN445 co-treatment was able to reverse the increased lipid uptake found in *ANGPTL4* KD ECs (Fig. 5f). In addition to the pharmacological inhibition of EL, we also targeted it with siRNA (Supplementary Fig. 8a–d). Interestingly, we also observed reduced *ANGPTL4* KD-mediated increase of lipid uptake from triolein when *LIPG* was co-silenced with *ANGPTL4* (Supplementary Fig. 8e). After the hydrolysis of TAG-containing lipoproteins, the resulting FAs are transported into ECs via several transport proteins[55]. We observed that KD of *ANGPTL4* increased the uptake of fluorescently labeled FA (BODIPY-FL- C16), which was reversed when cells were cultured in the presence of XEN445 (Fig. 5g). Additionally, increased surface expression of CD36 in *ANGPTL4* KD ECs was also normalized by XEN445 (Fig. 5h). Similar effects were observed on FA transport protein 4 (*FATP4)* expression (Supplementary Fig. 7c).

Next, we sought to determine the fate of the lipids that are taken up by ECs lacking ANGPTL4. Although ECs do not preferentially metabolize FA for ATP generation[56], FAO was significantly increased in ECs lacking *ANGPTL4* (Fig. 5i). Concomitantly, acetyl-CoA levels were also increased (Fig. 5j). The effects were circumvented with XEN455 treatment (Fig. 5i and j) or by *LIPG* co-silencing (Supplementary Fig. 8e), suggesting that the effect of *ANGPTL4* KD on FAO was mediated by its lipase inhibitory activity (Fig. 5k). In line with the increased FAO, we also found an increase in superoxide content (Supplementary Fig. 7d). We then validated mRNA and protein levels of molecules associated with FA uptake and metabolism. We observed an increase in mRNA or protein levels of CPT1A, and ACSL1 as well as CD36 total protein after *ANGPTL4* KD when compared to NS conditions (Supplementary Fig. 7e, f). Moreover, phosphorylated-AMPK was also increased, which is well-known in FA partitioning to regulate FAO[14] (Supplementary Fig. 7f). Interestingly, XEN445 or the co-silencing with *LIPG* normalized the levels of these molecules in *ANGPTL4* KD ECs (Supplementary Fig. 7e, f; Supplementary Fig. 8f, respectively). Besides enhanced FAO, we also sought to investigate additional fates of increased FAs in ECs with silenced *ANGPTL4*. ECs can store FAs in the form of lipid droplets, which protects them from FA-induced cellular stress[57]. We found an increased accumulation of neutral lipids (increased BODIPY staining) upon *ANGPTL4* KD (Supplementary Fig. 7g and Supplementary Fig. 8g) and increased mRNA levels of diacylglycerol acyltransferase 2 (*DGAT2*), an enzyme involved in cellular TG storage in the form of lipid droplets (Supplementary Fig. 7h). Inhibition or silencing of EL significantly attenuated these effects (Supplementary Fig. 7h and Supplementary Fig. 8g). Intriguingly, we did not observe increased ER stress genes or cell death when *ANGPTL4* was KD (Supplementary Fig. 7l and Supplementary Fig. 1c, d).

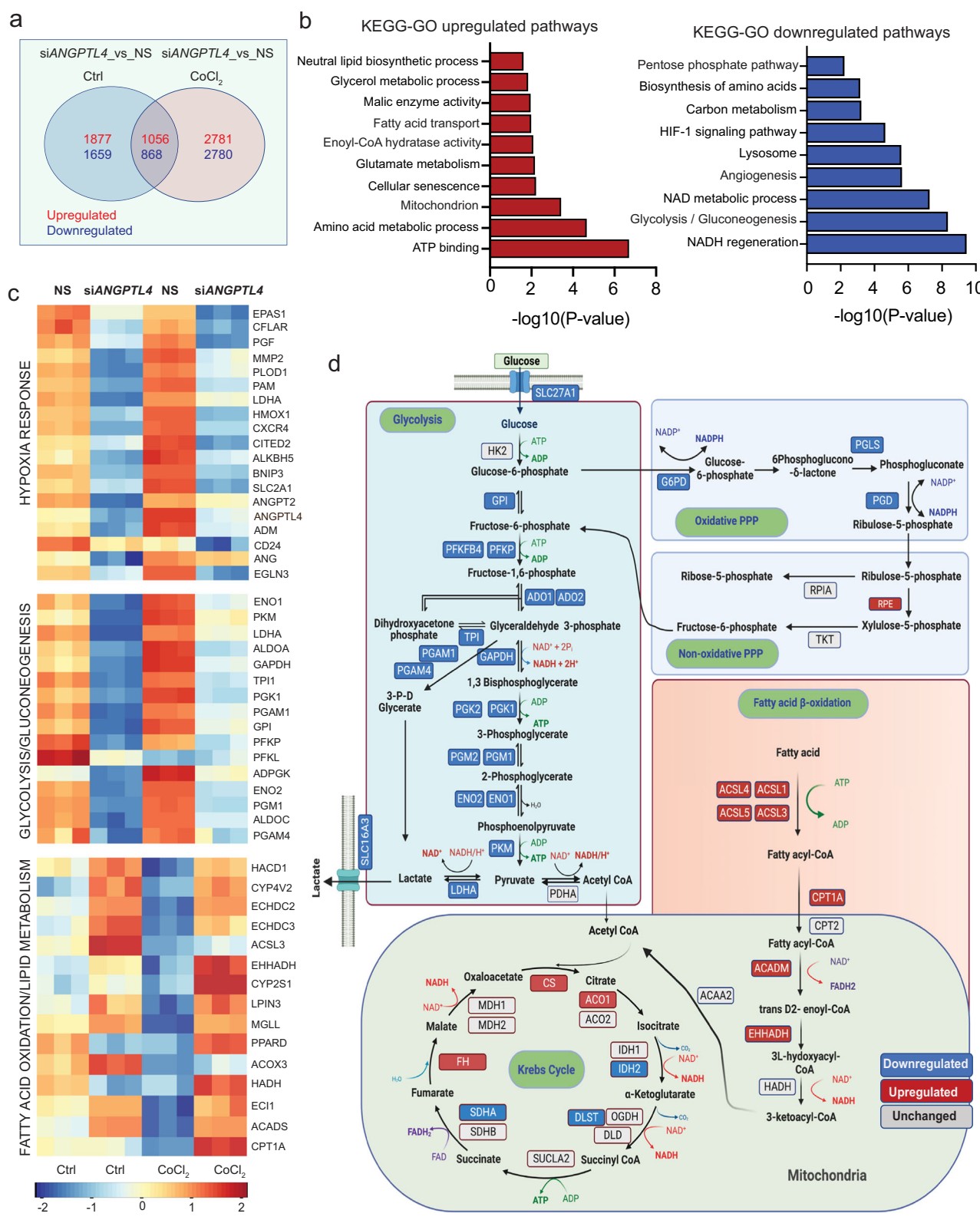

**Fig. 3 | Metabolic genes signature in HUVECs upon silencing *ANGPTL4* in the presence and absence of hypoxia mimetic CoCl₂.** HUVECs were transfected for 6 h with siRNA against *ANGPTL4* or NS control and then treated with CoCl₂ or vehicle control, cells were harvested 60 h post-transfection for RNA-seq analysis (*n* = 3 biological replicates). **a** Venn diagram showing the differentially expressed common genes between si*ANGPTL4* and NS treated with or without hypoxia mimetic CoCl₂. **b** KEGG-GO pathway analysis of 1056 differentially and significantly upregulated (red bars) and 868 downregulated (blue bars) genes in si*ANGPTL4* vs NS treated with CoCl₂ (significance for DGE was determined with Limma's deci-deTests). **c** Heatmap representing the transcript levels of indicated pathway genes. Color key: red, high correlation; blue, low correlation. **d** Pathway map representing changes in transcript levels of genes in central carbon metabolism in si*ANGPTL4* vs NS treated with CoCl₂. Color key: red, upregulated genes; blue: downregulated, gray: unchanged. Exact *p* values are shown for each comparison. Source data are provided as a Source data file.

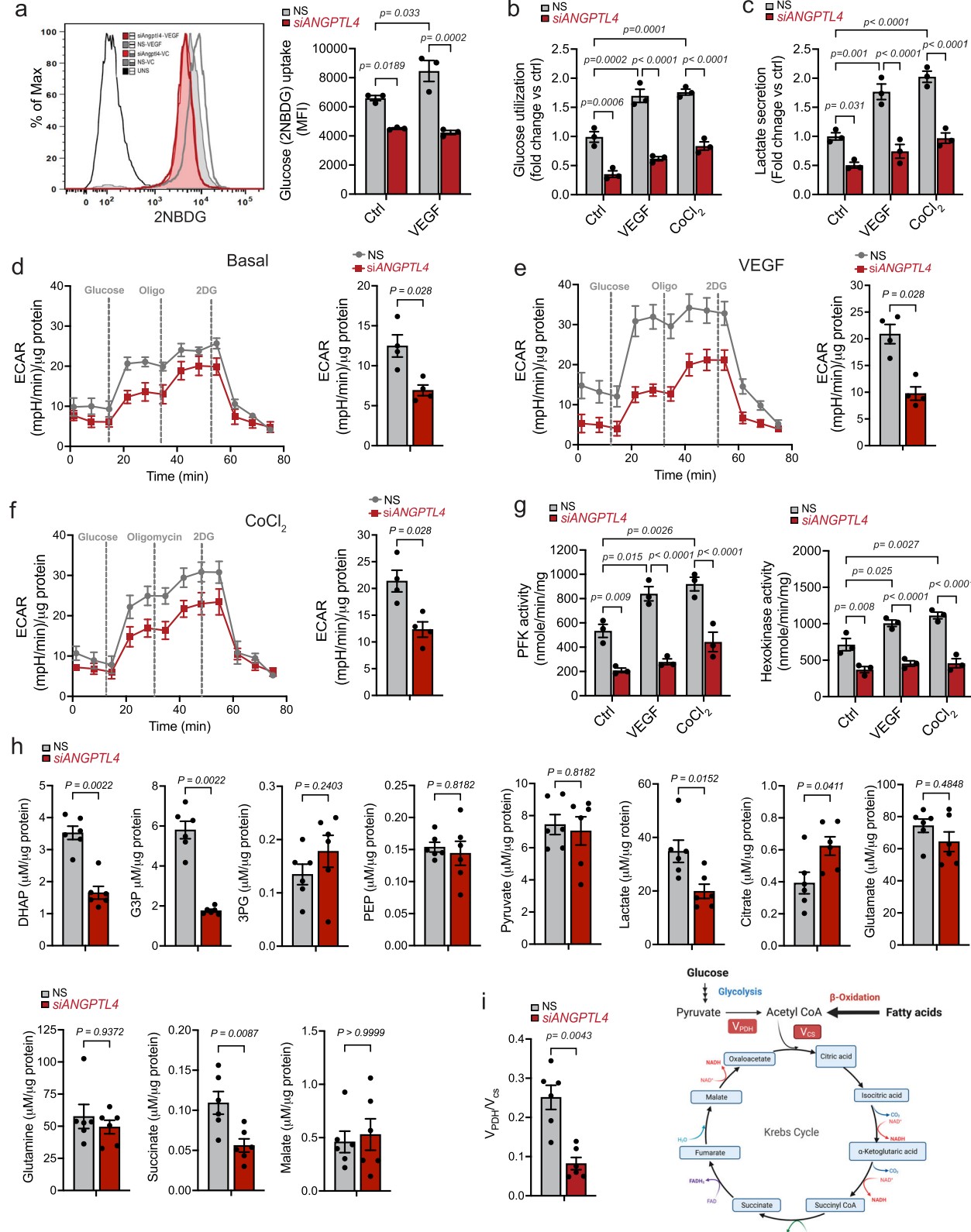

## EL inhibition partly reverses ANGPTL4 KD effects

To further validate whether lack of EL inhibition mediates the metabolic alterations due to *ANGPTL4* KD, we initially determined whether XEN455 treatment could restore the transcriptional changes induced by *ANGPTL4* depletion. We noticed that the expression of genes, altered after the silencing of *ANGPTL4*, was restored to a great extent by XEN445, as indicated by the number of genes that overlapped in

these conditions (Fig. 6a). KEGG-GO pathway and heatmap analysis of these overlapping transcripts indicated that EL inhibition mainly rescued gene signatures involved in regulating glucose, lipid metabolism, and angiogenesis (Fig. 6b and c). We then tested whether XEN445 or *LIPG* co-silencing could normalize the effect on glucose metabolism mediated by *ANGPTL4* KD in ECs. We found a significant increase in glucose uptake (Fig. 6d), PFK activity (Supplementary Fig. 9a),

**Fig. 4 | ANGPTL4 regulates EC glucose metabolism. a–c** HUVECs were transfected for 6 h with siRNA against ANGPTL4 or NS control and then treated with VEGF, CoCl$_2$ or vehicle control as indicated, cells were harvested 60 h post-transfection. **a** Glucose uptake determined by mean fluorescence intensity of 2NBDG via flow cytometry. **b** Glucose utilization determined as glucose remaining in the media after indicated incubation conditions. **c** Lactate secretion determined by measuring its levels in the media in indicated conditions. **a–c** (n = 3 independent experiments). **d–f** Seahorse analysis of glycolysis. HUVECs were transfected as described above. 48 h post-transfection, they were replated on Seahorse plates, incubated in basal media containing BSA (**d**), treated with VEGF (**e**) or CoCl$_2$ (**f**) for 24 h. Bar graphs on the left shows quantification of glycolysis rate (n = 4

independent experiment). **g** Activity of PFK and HK. Cells were incubated as in **a–c**. (n = 3 independent experiments performed in duplicate). **h, i** HUVECs were transfected as above, and cells were harvested 60 h post-transfection. **h** Steady-state levels of glycolysis and TCA cycle metabolites (Raw data are in Supplemental data 1). **i** MIMOSA analysis of metabolic flux through PDH and CS. Right panel shows the preference of substrate in the TCA cycle based on $V_{PDH}/V_{CS}$ data shown in left panel. **h, i** (n = 6 biological replicates). Data are represented as means ± SEM. Mann–Whitney U test in (**d–f, h, i**) and Two-way ANOVA with Tukey's multiple comparisons test in (**a–c, g**). Exact p values are shown for each comparison. Source data are provided as a Source data file.

extracellular proton flux (Fig. 6e and Supplementary Fig. 9b), NAD$^+$/NADH ratio (Supplementary Fig. 9c), and partial restoration of the mRNA and protein levels of key glycolytic enzymes (Supplementary Fig. 9d–g). These results suggest that elevated EL activity upon *ANGPTL4* KD promotes FA uptake and oxidation which might be a cause of suppressed glycolysis. To substantiate this, we measured ECAR in the presence of the FAO inhibitor etomoxir (Eto) and observed a partial rescue of the impaired glycolysis (Fig. 6f). Since XEN445 treatment upon *ANGPTL4* KD also partially rescued angiogenic gene signatures (Fig. 6c), we wonder whether this effect could also be attained at the functional level by improving the impaired angiogenic phenotype. Interestingly, lipase inhibition or its KD moderately restored the reduced EC proliferation and cord formation found in *ANGPTL4* KD ECs (Fig. 6g and Supplementary Fig. 9h).

### Lipase inhibition partly restores tumor angiogenesis

We next explored whether EC-specific deletion of *Angptl4* promotes tumor FA uptake. To do so, we collected LLC tumors 15 days post-implantation in WT and *Angptl4$^{iΔEC}$* mice. We first observed that ex-vivo lipase activity was increased in tumors isolated from *Angptl4$^{iΔEC}$* mice (Supplementary Fig. 10a). To measure intra-tumor lipid uptake, radio-labeled triolein was gavaged 2 h before sacrifice, which revealed increased lipid uptake in *Angptl4$^{iΔEC}$* tumors (Supplementary Fig. 10b). These results indicate that endothelial *Angptl4* KO elevates tumor-associated lipolysis and lipid uptake potentially via increasing local lipase activity. However, the uptake of lipids from triolein into adipose tissue (WAT) and liver was not altered in *Angptl4$^{iΔEC}$* mice when compared to WT (Supplementary Fig. 10c, d). In line with this, we did not observe alterations on circulating TAG, cholesterol, or HDL levels (Supplementary Fig. 10e). Taken together, these results suggest that ANGPTL4 in the quiescent vascular endothelium of WAT or liver does not significantly contribute overall lipase inhibition. However, in the angiogenic-activated tumor endothelium, ANGPTL4 might reduce lipolysis potentially inhibiting lipase activity in a local manner.

We then determined whether XEN445 treatment in *Angptl4$^{iΔEC}$* mice could normalize defects in EC function and tumor angiogenesis. *Angptl4$^{iΔEC}$* mice were subcutaneously injected with LLC cells. Once the tumor became palpable, mice were orally administered with XEN455 every other day, and tumor progression was monitored (Fig. 7a). While we did not observe a significant difference in tumor growth between the WT mice receiving either vehicle control (DMSO) or XEN445, we did notice that XEN445 administration to *Angptl4$^{iΔEC}$* mice enhanced tumor growth as compared to *Angptl4$^{iΔEC}$* mice receiving vehicle control (Fig. 7b). XEN445 partially reversed the tumor angiogenic defects observed in *Angptl4$^{iΔEC}$* mice (Fig. 7c). XEN445 also significantly reversed tumor-induced increased vascular permeability observed in *Angptl4$^{iΔEC}$* mice, assessed as intra-tumoral hemorrhage analysis by Ter119 staining (Supplementary Fig. 11a), with no effect of XEN445 in WT mice (Supplementary Fig. 11a). We additionally tested whether XEN445 administration could normalize the glucose uptake defect in tumor ECs in vivo by intravenously injecting the fluorescent glucose analog 2NBDG in tumor-bearing mice and determining the colocalization of fluorescent glucose within CD31$^+$ ECs. Tumors from

*Angptl4$^{iΔEC}$* mice had an overall reduced uptake of 2NBDG (Fig. 7d) that was accompanied with reduced colocalization within ECs. Notably, XEN445 administration rescued this effect and increased EC glucose uptake in tumors from *Angptl4$^{iΔEC}$* mice (Fig. 7d). These results highlight a potential crosstalk of EC metabolic pathways in vivo. Next, we analyzed FA uptake and TAG lipolysis in ECs in *Angptl4$^{iΔEC}$* mice since the defect in glucose metabolism in vitro was primarily associated with elevated FA uptake and oxidation upon *ANGPTL4* KD. First, mice were orally administered with fluorescently labeled FA, which would then be esterified, packaged within chylomicrons in the gut, remodeled, and transported to different tissues (Supplementary Fig. 11b). We observed enhanced FA uptake in the ECs and non-EC parenchymal cells from *Angptl4$^{iΔEC}$* tumor-bearing mice as compared to WT tumor-bearing mice, and administration of either the EL inhibitor XEN445 or the pan-lipase inhibitor GSK264220A in *Angptl4$^{iΔEC}$* mice reduced lipoprotein-derived labelled FA uptake in tumor ECs (Fig. 7e and Supplementary Fig. 11c–e). Interestingly, and in line with the results, depletion of EC *Angptl4* increased endothelial CD36 levels (Fig. 7f). Thus, increased lipase activity within the tumor and elevated CD36 expression favors lipid mobilization and utilization in the tumor-EC compartment when endothelial *Angptl4* is absent.

Our results indicate that an increase in lipase activity within the tumor is responsible for the anti-tumor and anti-angiogenic effects mediated by endothelial ANGPTL4 deficiency and suggest that proper regulation of local lipase activity in ECs is important for the regulation endothelial functions and tumor angiogenesis.

## Discussion

The paracrine role of ANGPTL4 in ECs in blood vessel formation and integrity has been studied with inconclusive results. However, the specific role of endothelial ANGPTL4 in regulating EC functions and metabolism is largely unknown, especially regarding endothelial bioenergetics and angiogenesis. Here we report that endogenous EC ANGPTL4 promotes a metabolic phenotype that favors angiogenesis. Briefly, we identified that, by regulating local lipase activity in ECs, ANGPTL4 controls FA utilization to sustain the use of glucose for angiogenesis.

ANGPTL4 is a multifunctional protein whose expression and function are greatly influenced by microenvironmental and patho-physiological conditions[58] suggesting a context-, cell type- and tissue-specific activity. Ample literature unequivocally involves ANGPTL4 in regulating lipid metabolism[2,3,59,60]. However, its role in regulating two directly related processes: angiogenesis and vascular permeability, has also been heavily documented with conflicting results[2], either repressing vascular permeability and angiogenesis[18,21,22,28,30,61,62], or having proangiogenic actions and destabilizing barrier functions[17,19,29,32,63–65]. However, most of these studies primarily involved exposure of endogenous ANGPTL4-expressing ECs to exogenous ANGPTL4 or exploiting whole-body KO mice, thus neglecting a potential EC-specific autonomous/autocrine role of ANGPTL4. In the present work, we found that suppressing *ANGPTL4* expression in ECs reduces their angiogenic functions, which supports a pro-angiogenic role for endothelial ANGPTL4. We found that this effect was accompanied by

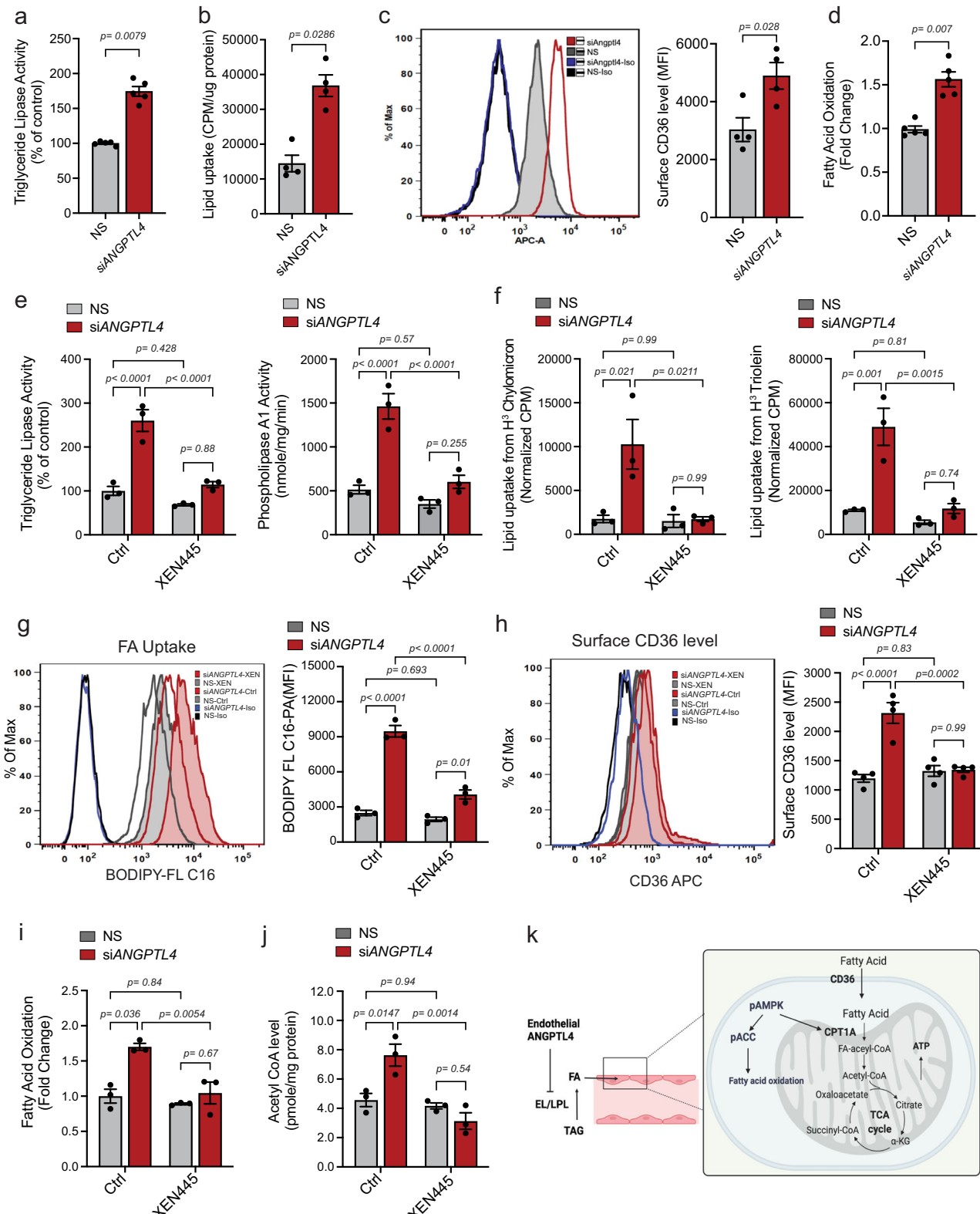

transcriptional changes in gene signatures associated with angiogenic processes such as cell adhesion, migration, and morphogenesis. Furthermore, pro-angiogenic gene expression signatures induced by hypoxia[66] were also downregulated upon *ANGPTL4* silencing in ECs, which is in line with initial observations that *ANGPTL4* expression was induced by hypoxia in human ECs to mediate a strong proangiogenic response in the chicken chorioallantois membrane assay[17]. In vivo, in contrast to previous work showing decreased perinatal retinal

angiogenesis when *Angptl4* was deleted in a global manner[32], we found that endothelial KO of *Angptl4* in neonatal mice did not affect physiological retinal angiogenesis while affecting pathological tumor-induced angiogenesis. The effect on pathological tumor-induced angiogenesis occurred in the absence or presence of tumor-derived *Angptl4*, which in turn has been described to have both pro- or anti-angiogenic effects[47], indicating that endothelial ANGPTL4 facilitates pathological angiogenesis cell-autonomously, since reduced tumor

**Fig. 5 | ECs lacking ANGPTL4 exhibit elevated lipoprotein lipolysis, FA uptake and metabolism. a–d** HUVECs were transfected for 6 h with siRNA against *ANGPTL4* or NS control, cells were harvested 60 h post-transfection **a** Triglyceride lipase activity in the heparinized culture medium (*n* = 5 independent experiments). **b** Lipid uptake from [³H]-triolein **c**, representative histograms for CD36 surface protein levels determined by flow cytometry, M.F.I was used for quantification on the right. **b, c** (*n* = 4 independent experiments). **d** FAO determined by oxidation of ¹⁴C-Palmitate (*n* = 5 independent experiments). **e** HUVECs were transfected as above, 60 h post-transfection heparinized media was collected and incubated for 1 h with or without XEN445 at 37 °C, then triglyceride lipase and phospholipase activity was measured. **f–k** HUVECs were transfected as above and then treated with

XEN445 or vehicle control, cells were harvested 60 h post-transfection. **f** Lipid uptake from [³H]-Chylomicron (left panel) and [³H]-triolein (right panel). **g** Representative histograms for (BODIPY-FL-C16) uptake determined by flow cytometry, M.F.I was used for quantification of on the right. **h** Surface protein level of CD36 as in **c. i** FAO determined by oxidation of C¹⁴-Palmitate. **j** Level of cellular Acetyl CoA. (**e–j**, *n* = 3 independent experiments). **k** Schematic diagram showing the mechanism of ANGPTL4 mediated EL inhibition and FA uptake and metabolism in ECs. data are represented as means ± SEM. Two-way ANOVA with Tukey's multiple comparisons test in (**a–j**). Exact *p* values are shown for each comparison. Source data are provided as a Source data file.

angiogenesis observed in *Angptl4*^*iΔEC*^ mice could not be compensated by ANGPTL4 derived from cancer cells that also express and produce ANGPTL4[4,31,47]. In contrast to previous studies that evaluated the paracrine actions of ANGPTL4[21,22], our results further emphasize that EC-derived ANGPTL4 is required to support pathological angiogenesis. Although physiological and pathological angiogenesis share many similarities, some distinctions between these two events are overt, implying subtle differences in underlying mechanisms. The interplay of EC ANGPTL4 with the described regulators of angiogenesis in physiological settings vs. pathological settings could be different – an area that warrants future investigation.

In addition to its impact on tumor-induced angiogenesis, we found that EC-specific deletion of *Angptl4* decreased tumor-mediated vascular permeability. ANGPTL4 has been associated with tumor metastasis by modulating the integrity of vascular EC layers[21,31]. It was described that ANGPTL4 affects vascular permeability through integrin αvβ1-mediated PAK/Rac signaling, weakening EC–EC contacts[67]. In another setting, ANGPTL4 induced vascular permeability by binding to neuropilin 1 (NRP1) and NRP2, activating the RhoA/ROCK signaling pathway and leading to the breakdown of EC junctions[65]. Alternatively, ANGPTL4 has been reported to prevent the metastatic process by inhibiting vascular permeability through αvβ3 interaction, thereby enhancing adherens and tight junction integrity and promoting pericyte recruitment[22]. Our present findings show that tumor vessels in *Angptl4*^*iΔEC*^ mice are less permeable and have increased pericyte coverage, potentially reducing tumor vascular leakage. In vitro, we found decreased mRNA and surface expression of αvβ3, and to a lesser extent of αvβ5, in the absence of ANGPTL4 in ECs, suggesting that different mechanisms, than those described previously, might be at play in *ANGPTL4* KD ECs. In this regard, we also found that ECs lacking *ANGPTL4* upregulate FoxO and Hippo pathways, known to reduce angiogenesis and promote EC quiescence and vascular stability[39–41]. These changes are linked to modifications in EC metabolism changes are linked to modifications in EC metabolism[53], supporting our hypothesis that ANGPTL4-deficient ECs activate safeguard mechanisms to maintain cell viability and their quiescent state.

While homeostatically quiescent, ECs become activated during angiogenesis, thus increasing their metabolic demand often distinctively from that of their resident tissue[34]. Recent insights suggest that changes in EC metabolism can play a significant role, driving angiogenesis alongside established angiogenic growth factors[24,68,69]. As a result, targeting EC metabolism has gained attention as a potential therapeutic approach to inhibit angiogenesis. One interesting discovery in our study is that the absence of endothelial ANGPTL4 not only downregulates genes associated with angiogenic responses but also disrupts relevant EC metabolic pathways. Notably, glycolytic gene signatures, a well-known metabolic pathway during angiogenesis[56], were downregulated when *ANGPTL4* was silenced in ECs. Simultaneously, we observed an increase in gene signatures related to lipid metabolism, particularly in the utilization of fatty acids, which have been linked to quiescence[42,70]. Intriguingly, we also observed an upregulation of FOXO family genes, known to regulate c-Myc and its downstream metabolic genes[40] and Hippo signaling pathways,

particularly the downregulation of transcriptional co-activators YAP1/TAZ, well-known to regulate metabolic processes and angiogenesis[40–42].

In addition to LPL and HL, ANGPTL4 also inhibits EL[52]. Due to the presence of EL and GPIHBP1-anchored LPL on the luminal surface, ECs also participate in vascular lipolysis, FA uptake, and trans-endothelial transport to parenchymal tissues[33,34]. However, the effect of EC ANGPTL4-mediated regulation of lipolysis on EC functions and angiogenesis remains unexplored. Our findings show that in the absence of ANGPTL4, EL activation increased lipid uptake and oxidation while reducing glycolysis and angiogenic responses in vitro. This might be due to excess citrate generated from FAO, which can inhibit the rate-limiting glycolytic enzyme PFK[53]. Our metabolomics data align with this, indicating higher citrate levels in ECs lacking ANGPTL4, associated with reduced PFK activity, impaired glycolysis, and defective angiogenic responses. Our results are consistent with previous studies showing that reduced EC glycolysis can decrease angiogenic responses and reduce metastasis, at least in part by promoting tumor vessel normalization[50,71]. Increased free fatty acids (FFA) have been reported to reduce EC activation and angiogenesis[42,70]. On the other hand, absence of endothelial CPT1α has been described to cause vascular sprouting defects due to reduced proliferation without affecting migratory responses. Furthermore, inhibition of FAO, using etomoxir, led to similar results, while overexpression of CPT1α produced the converse effects[72]. The primary distinction with our current findings lies in the forced increase in FAO caused by the enhanced FA uptake resulting from increased lipolysis in the absence of ANGPTL4. This surge of FA uptake and oxidation counteracts glycolysis, potentially affecting both the proliferative and migratory responses of EC[50]. However, the report of Schoors et al. is related to basal FAO-derived carbons for angiogenesis. In their setting, there were no changes in FA uptake, while glycolysis was not altered[72]. RNA-seq data revealed that XEN445 partially rescued glycolysis-related gene signature and normalized lipid and FA metabolic gene signatures in ECs lacking ANGPTL4. This effect was accompanied with a partial rescue of the angiogenic gene signature and the capacity to utilize glucose, reflecting partial normalization of angiogenesis both in vitro and in vivo. Although it's unclear if FAs directly influence the observed gene expression changes, they might exert their effects indirectly via FOXO or YAP1/TAZ pathways. Alternatively, increased FFA in ECs may act as peroxisome proliferator-activated receptor (PPARα) ligands, potentially contributing to maintaining quiescent, non-activated states, as seen in inflammatory activation[73].

Together, our data suggest that EL-mediated enhanced lipolysis and FA uptake modulates EC metabolism. Normalizing EC metabolism after *ANGPTL4* KD rescued to some extent the angiogenic defects observed in vitro, highlighting how EC-derived ANGPTL4 maintains the cellular metabolic homeostasis required for angiogenesis. However, it remains unclear whether ANGPTL4 also regulates angiogenic function in ECs by binding to a specific receptor to exhibit its downstream signaling, or as an intracellular modulator of different signaling cascades. Alternatively, whether these actions are regulated in conjunction with our newly described metabolic effects.

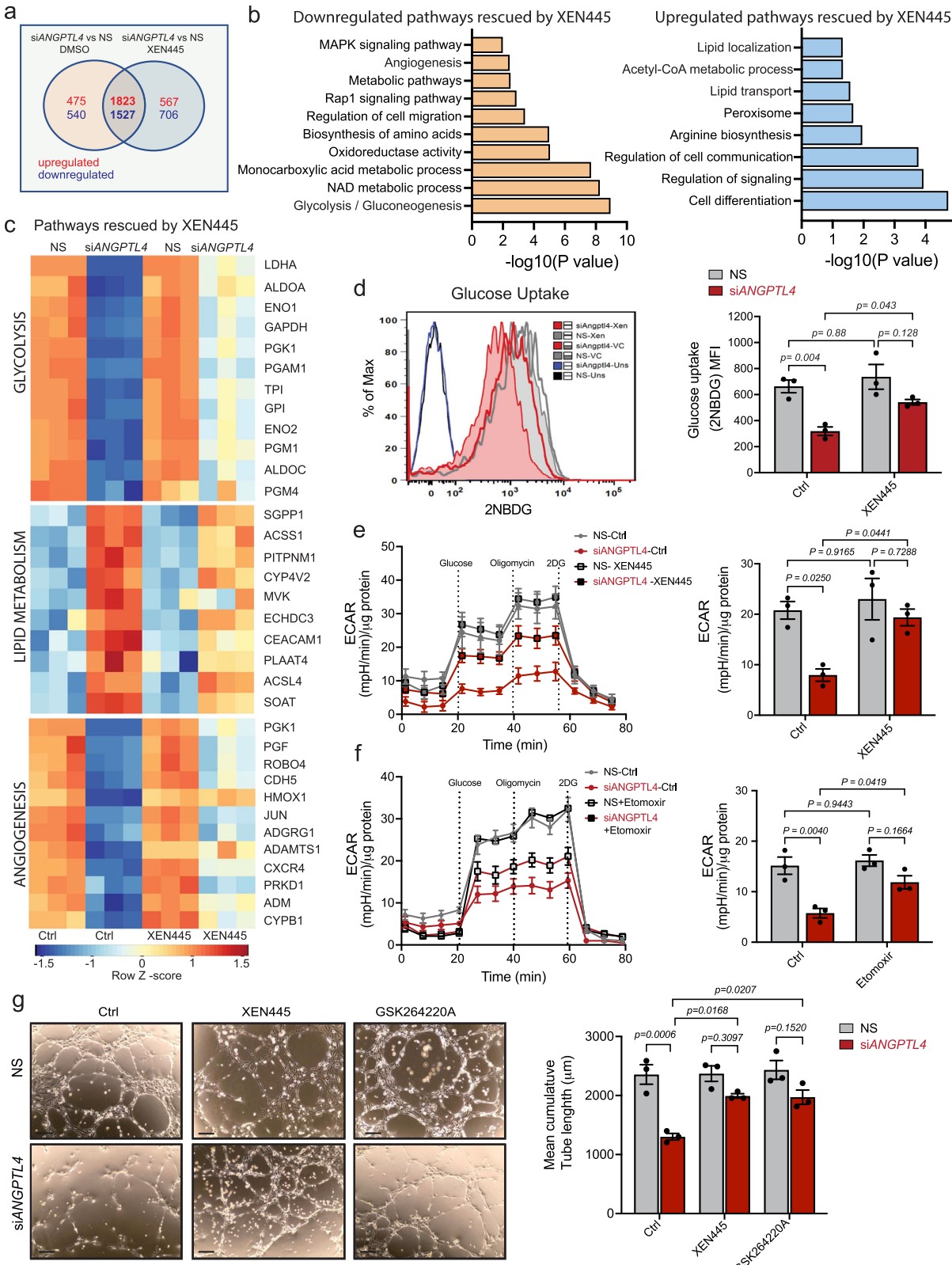

**a** siANGPTL4 vs NS DMSO / siANGPTL4 vs NS XEN445. 475, **1823**, 567 upregulated; 540, **1527**, 706 downregulated.

**b** Downregulated pathways rescued by XEN445 / Upregulated pathways rescued by XEN445. -log10(P value)

**c** Pathways rescued by XEN445. GLYCOLYSIS, LIPID METABOLISM, ANGIOGENESIS. Row Z-score

**d** Glucose Uptake. 2NBDG. Glucose uptake (2NBDG) MFI.

**e** ECAR (mpH/min)/µg protein. Glucose, Oligomycin, 2DG. Time (min).

**f** ECAR (mpH/min)/µg protein. Glucose, Oligomycin, 2DG. Time (min).

**g** Ctrl, XEN445, GSK264220A. NS, siANGPTL4. Mean cumulative Tube lenghth (µm)

Recent studies have shown that ECs not only transport FAs to parenchymal cells but also use them for ATP generation and biomass production[35,36,74]. Following lipase-dependent TG hydrolysis, FAs are taken up by ECs via different FA transporters including CD36[35–37]. Interestingly, prior work has an inverse correlation between the expression of ANGPTL4 and CD36[75], whereby CD36 in EC acts as a gatekeeper for parenchymal cell FA uptake[35]. Excess FA retention in EC

might affect the EC function, particularly glycolytic metabolism and subsequent angiogenic processes[42,70], and increased CD36 expression is correlated with decreased angiogenesis[76]. Consistent with these findings, we noticed increased FA uptake and retention in cancerous cells as well as in tumor-associated ECs of *Angptl4*[iΔEC] mice. Moreover, inhibition of lipase activity by XEN445 or GSK264220A decreased lipid uptake and normalized the level of CD36 in ECs, suggesting that the

**Fig. 6 | EC specific suppression of *ANGPTL4* expression rewire glucose metabolism in EL dependent manner. a–c** HUVECs were transfected for 6 h with siRNA against *ANGPTL4* or NS control, then treated with XEN445 or vehicle control. Cells were harvested 60 h post-transfection for RNA-Seq analysis (*n* = 3 biological replicates). **a** Venn diagram showing the differentially expressed common genes between si*ANGPTL4* and NS treated with or without XEN445. **b** KEGG-GO pathway analysis of 1823 differentially and significantly upregulated and 1527 downregulated genes in si*ANGPTL4* vs NS treated with XEN445 (significance for DGE was determined with Limma's decideTests). **c** Heatmap analysis of top pathways (glycolysis, lipid metabolism and angiogenesis) rescued by XEN445 upon silencing *ANGPTL4*. Color scale: red, high correlation; blue, low correlation. **d** Glucose (2NBDG) uptake as in Fig. 4a in cells incubated as above in the presence of XEN445 (*n* = 3 independent experiment performed in duplicate). **e** and **f** Glycolysis analyzed by

Seahorse. HUVECs were transfected and treated as described above. At 48 h post-transfection, cells were replated onto Seahorse assay plates and incubated either with or without XEN445 for an additional 24 h (**e**) or with etomoxir for an additional 12 h (**f**) prior to Seahorse analysis to measure the ECAR. (**e** and **f**) right panel shows the quantification of glycolytic rate (ECAR) (*n* = 3 independent experiments). **g** HUVECs were transfected as above, then treated with XEN445 or GSK264220A and harvested 60 h post-transfection for Matrigel cord formation assay as in Fig. 1k in basal conditions. Representative micrographs (left panel) and quantification (right panel) of cumulative tube lengths. (*n* = 3 independent experiments performed in duplicate). Scale bars, 50 µm (**g**). Data are represented as means ± SEM. Two-way ANOVA with Tukey's multiple comparisons test in (**d**–**g**). Exact *p* values are shown for each comparison. Source data are provided as a Source data file.

effect of ANGPTL4 on lipase activity is, at least in part, responsible for alterations in CD36 levels, though the exact mechanisms require further investigation.

Importantly, anti-angiogenic signaling has been shown to promote FA uptake in ECs by increasing the expression of *CD36* and *LIPG* and negatively regulating the expression of *ANGPTL4*[75], suggesting that anti-angiogenic signaling can also exert its effects by metabolic regulation[24]. Interestingly, we observed that depletion of ANGPTL4 in ECs reduces the expression of VEGFR2 and diminishes VEGF-mediated activation of ERK1/2 signaling and downstream angiogenic responses. On the contrary, another study that evaluated the paracrine effect of ANGPTL4, on EC expressing ANGPTL4, showed that ANGPTL4 negatively regulated P-Y1175-VEGFR2/ERK1-2 signaling impacting EC-cell junction patterning[61]. Nevertheless, our data suggest that EC-derived ANGPTL4 may exert its pro-angiogenic function in an autocrine cell-autonomous manner by inhibiting local lipase activity. However, the absence of mouse-specific antibodies hinders a complete understanding of how different secreted forms of ANGPTL4 may interact with endothelial ANGPTL4 in regulating angiogenic responses, which could help reconcile previous work examining paracrine actions of different ANGPTL4 forms in EC functions.

In cell-based activity assays, ANGPTL4 has been shown to inhibit EL[52]. However, genetic mimicry analysis in humans does not support EL inhibition by ANGPTL4 in regulating circulating lipids[77]. Our in vitro studies support the idea that ANGPTL4's effects on EC metabolism and angiogenesis are largely mediated through its inhibition of EL, which is highly expressed in human ECs. Nevertheless, the absence of endothelial ANGPTL4 may enhance tumoral lipid uptake, potentially due to increased lipase activity in the tissue. It's essential to consider that some of these effects might also be mediated by the lack of LPL inhibition within the activated vasculature of the tumor, as a pan-lipase inhibitor also reduced FA uptake in tumor-associated ECs. Interestingly, we found no effect on lipid uptake from triolein in WAT or the liver, suggesting that the contribution of endothelial ANGPTL4 to lipase inhibition (LPL or EL) in these tissues, that are not under angiogenic activation, could be considered residual when compared to ANGPTL4 from adipocytes and hepatocytes[12,14,15,78,79]. Accordingly, we found that circulating lipid levels were not affected in *Angptl4*[iΔEC] mice. While EL has been known to regulate VLDL and LDL metabolism in both humans and mice, it primarily influences HDL metabolism. Thus, one would expect that the absence of endothelial ANGPTL4 would affect HDL levels to some extent. However, studies in mice have shown inconsistent results regarding the impact of ANGPTL4 on HDL-C[80]. Plasma HDL levels are not altered by ANGPTL4 deletion in mice. Furthermore, human carriers of ANGPTL4 E40K showed increased plasma HDL levels rather than decreased[81]. Since our experiments were performed in animals under chow feeding, it remains to be investigated the participation of endothelial ANGPTL4 in metabolic pathophysiological conditions such as adipose tissue expansion under high fat diet feeding that requires an angiogenic activated endothelium[82].

Previously, it has been demonstrated that activation of *LIPG* in breast cancer cells promotes extracellular lipid uptake and facilitates tumor growth and progression[83]. In a similar line, we also observed increased expression of *LIPG* and corresponding lipase activity after silencing of *ANGPTL4* in ECs, but in our case, we found reduced angiogenesis and tumor growth. On the other hand, *LPL* gene deficiency increases cancer risk. Interestingly, tumor-suppressive effects of LPL inducers, such as PPAR ligands, NO-1886, and indomethacin, have been demonstrated in animal models[84].

Altogether, our studies indicate that in pathological angiogenic conditions, ANGPTL4 is stimulated in ECs to maintain local lipase inhibition, thus diminishing FA utilization and activating a glycolytic state that supports angiogenesis. Our study highlights the importance of maintaining the balance between FA and glucose metabolism for EC quiescence and angiogenesis[39,40,68]. It is nowadays increasingly accepted that manipulation of EC metabolic pathways alone (even without changing angiogenic signaling) alters angiogenic responses and can underlie the excess formation of new blood vessels in tumors[24]. Our results indicate that EC-derived ANGPTL4 is required for maintaining metabolic homeostasis, not only regarding lipid uptake and transport but also regarding intracellular EC bioenergetics for neovascularization. Our study opens the door to exploring endothelial ANGPTL4 as an EC-metabolism-focused therapeutic avenue to target pathological angiogenesis.

## Methods
### Mice
Experiments were conducted under the ethical guidelines and protocols approved by IACUC (Institutional Animal Care and Usage Committee) in Yale University School of Medicine (Animal protocol, 2022-116576). To generate inducible endothelial-specific *Angptl4*–deficient mice, *Angptl4*[flox/flox] mice[12] were crossed to *Cadh5-Cre*[ERT2] mice[85] to achieve specific inactivation after tamoxifen (TMX) Cre mediated excision (referred to as *Angptl4*[iΔEC] mice). All mouse strains were in the C57BL6J genetic background. *Angptl4*[flox/flox] mice (hereafter referred to as WT mice) are also injected with TMX. For analysis at postnatal day (P) 6 neonates, 50 µg of TMX was injected i.p. on three consecutive days, P1-P3 as previously described[86]. For adult angiogenesis/tumor progression studies, *Angptl4* KO was induced in EC via 5 consecutive i.p. daily injections of tamoxifen (100 mg/kg) to 4-week-old mice[86]. Genotype of *Angptl4*[flox/flox] mice was verified by PCR using ERT2 primers and primers flanking the 5' homology arm of the *Angptl4* gene and LoxP sites from the tail-extracted DNA. KO efficiency was verified in endothelial cells by sorting different cell populations (CD45⁻CD31⁺-ECs; CD45⁺ CD31⁻ -lymphocytes, CD45⁻CD31⁻ - parenchymal cells) followed by RT-qPCR, using primers that expand exon 5 (Primers sequences in Supplementary Table 1). All mice were housed in a barrier animal facility with a constant temperature and humidity in a 12-h dark/light cycle. All mice were fed with a standard chow diet (CD) and water and food were provided *ad libitum*. Both male and female mice were used since no sex-specific differences were observed.

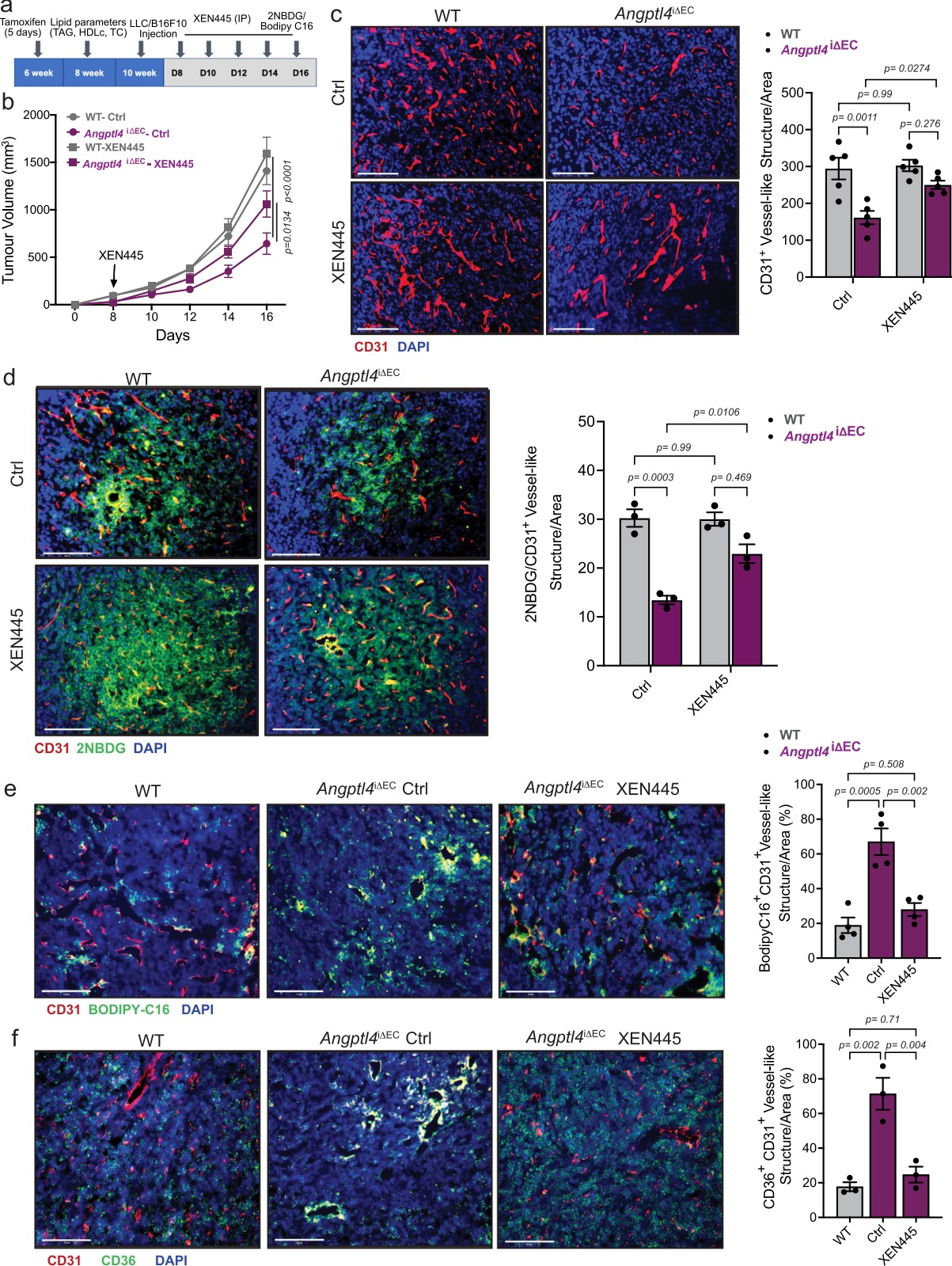

## Cell culture and treatments

HUVECs were obtained from the tissue culture core laboratory of the Vascular Biology and Therapeutics Program (Yale University, New Haven, CT) and cultured in M199/EBSS medium supplemented with 20% FBS) and EC growth factor (ECGS, 1:200) (Gibco), as described[86]. When required, cells were treated with lipase inhibitor XEN445 (10μM)

(Cyman Chemicals) or GSK264220A (10μM) (Tocris) for indicated time points. Equivalent volume of DMSO was used as a vehicle control. In some instances, cells were treated with 50 ng/ml VEGFA (R&D Systems, Cat# 293-VE-010/CF), 100 μM CoCl2 or 50 μM Etomoxir (Sigma) for indicated time points. LLC (Cat# CRL-1642) and B16F10 (Cat# CRL-6475) cells were purchased from ATCC and were used for tumor

**Fig. 7 | Inhibition of EL activity in EC-specific *Angptl4* KO mice normalizes EC metabolism and angiogenesis. a** Scheme showing the experimental strategy. **b**–**f** Tumor analysis of WT Ctrl (*n* = 5), WT XEN445 treated (*n* = 6), *Angptl4*[iΔEC] Ctrl (*n* = 5) mice and *Angptl4*[iΔEC] XEN445 treated (*n* = 6) with s.c. injection of LLCs in the dorsal flank. **b** Tumor volume. **c** Left, representative micrographs of CD31 (red) and DAPI (blue) immunostaining. Right, Quantification of CD31[+] vessel-like structures (*n* = 5 quantified) **d** Left panel, representative micrographs of immunofluorescence staining of CD31 (red), 2NBDG (green) and DAPI (blue). Right panel, quantification of 2NBDG positive CD31 positive vessel-like structures. **e** Left, Representative

micrographs showing immunostaining of CD31 (red) and fluorescent FA (BODIPY-FL-C16) (green). **f** Left, representative micrographs of CD31 (red) and CD36 (green) immunostaining. Right panels (**d**–**f**) are quantification of at least 4 different images from each mouse. (*n* = 3 quantified per genotype out of 5 or 6 randomly selected, except for **e** (*n* = 4 quantified). Scale bars, 100μm (**c** and **d**), 125 μm (**e** and **f**). Data are represented as means ± SEM. Two-way ANOVA with Tukey's multiple comparisons test in (**b**–**d**). One-way ANOVA with Tukey's multiple comparisons test in (**e**) and (**f**). Exact *p* values are shown for each comparison. Source data are provided as a Source data file.

growth studies, were cultured in DMEM (Corning), with 10% FBS (Corning), 1% GlutaMAX (Gibco), and 1% antibiotic/antimycotic (Corning).

## Transfection with siRNAs

HUVECs (2 ×10[5]/well) were plated in 6-well plates in above indicated media. Cells were allowed to adhere overnight. siRNA-mediated knockdown of ANGPTL4 and/or LIPG was performed using sets of four SMARTpool siRNAs (Dharmacon). Transfection was carried out using RNAiMAX (Life Technologies). Briefly, prior to transfection, the medium was replaced with fresh, serum-reduced Opti-MEM. The cells were incubated in this transfection medium (OptiMEM containing siRNA and RNAiMAX) for 6 h. Afterwards, medium was switched back to the standard HUVEC culture medium containing 20% FBS. Cells were then incubated for varying durations, ranging from 24 to 96 h, although most experiments were conducted 60 h post-transfection (highly efficient gene silencing). Non-silencing (NS) siRNAs were used as control. Knockdown efficiency was evaluated by qRT-PCR, ELISA or immunoblot. siRNA information in Supplementary Table 2.

## Generation of Tumor cell line depleted of ANGPTL4

Stable knockdown for *Angptl4* was obtained by lentiviral mediated transduction of specific shRNAs against *Angptl4* (Horizone Discovery, Supplementary Table 2) in LLC cells and selected in puromycin[87]. Knockdown efficiency of *Angptl4* was evaluated by qRT-PCR for *Angptl4*.

## ANGPTL4 elisa

ANGPTL4 levels in HUVECs culture media or in cell lysate were determined using a commercially available ELISA kit (Human Angiopoietin-like 4 DuoSet ELISA, R&D Systems, Cat #DY3485). At the indicated post-transfection times, medium was collected, centrifuged to remove floating cells, and stored at 4 °C. Cells were washed with PBS and lysed in lysis buffer (see below) containing protease inhibitors and stored at −80 °C. 100 μL of undiluted medium or 25 μL cell lysate were used respectively for ELISA detection. Optical density (OD) was measured with a SpectraMax microplate reader (Molecular Devices) with absorbance set at 450 nM and 540 nM. ANGPTL4 levels were normalized total protein content and expressed as ng/ml/mg protein.

## Collection of ANGPTL4 enriched and depleted conditioned medium

To obtain conditioned media from HUVECs enriched or depleted of ANGPTL4 (hypoxia-induced in NS Ctrl HUVECs or hypoxia-induced siANGPTL4 HUVECs, respectively), cells were transfected with either NS or siANGPTL4 as above indicated. After transfection, cells were incubated in a hypoxic incubator set at 1% $O_2$ balanced with $N_2$ for 48 h. Media was collected and stored. Enrichment of ANGPTL4 was determined by ELISA.

## RNA isolation and quantitative real-time PCR

Total RNA from transfected cells or from the tissues was isolated using TRIzol reagent (Invitrogen) according to the manufacturer's protocol. cDNA was prepared using iScript RT Supermix (Bio-Rad), as per manufacturer's instruction. Quantitative real-time PCR (qRT-PCR) was

performed using EvaGreen Supermix (BIORad) on an iCycler Real-Time Detection System (BIORAD). The mRNA levels were normalized to 18 S rRNA. Primers sequences in Supplementary Table 1.

## RNA-sequencing

Total RNA was isolated from HUVECs that were transfected with NS or siANGPTL4 and treated with or without $CoCl_2$ or lipase inhibitor XEN445, using Qiagen's RNeasy kit. The purity and integrity of the RNA samples was analyzed using a Bioanalyzer (Agilent Technologies) at Yale Center for Genomic Analysis (YCGA). RNA (500 ng) was purified with oligo-dT beads and cDNA library is end-repaired, and A-tailed using a Kapa mRNA Hyper prep kit (Cat# KK8581, Wilmington, MA). Indexed libraries quantified by qRT-PCR using a commercially available kit from Kapa DI Adapter Kit (Cat# KK8722, Wilmington, MA) and insert size distribution determined with the LabChip GX or Agilent Bioanalyzer. Samples are sequenced using 100 bp single or paired-end sequencing on an Illumina HiSeq 2500 according to Illumina protocols. Primary analysis - sample de-multiplexing and alignment to the human genome - was performed using Illumina's CASAVA 1.8.2 software suite.

## Analysis of RNA-sequencing data and pathway analysis

Paired-end reads were imported into Partek Flow (Copyright ©; 2018 Partek Inc., St. Louis, MO, USA) and validated to have an average read quality >35. Reads were aligned to the *Homo sapiens* genome assembly GRCh38 (hg38) with the STAR aligner and annotated with Ensembl in Partek Flow software. Raw FASTQ files and processed data (aligned and annotated raw gene counts) are available at Gene Expression Omnibus (GSE211128). After post-alignment QA/QC validation that percentage of aligned reads was >90%, total counts per gene were quantified and resultant gene count table imported into R for further analysis. Genes with more than ~50 counts in at least three samples were included, and subsequent quality control and differential analysis were performed with edgeR-limma-voom as described (https://www.ncbi.nlm.nih.gov/pmc/articles/PMC4937821/, https://genomebiology.biomedcentral.com/articles/10.1186/gb-2014-15-2-r29). For downstream analysis, significance was determined with Limma's decideTests function with a Log2 Fold Change of greater than 0.5 and a Benjamini-Hochberg False Discovery Rate of less than 0.05. Pathway analysis, both KEGG (https://doi.org/10.1093/nar/28.1.27, https://doi.org/10.1093/nar/gkaa970, https://doi.org/10.1002/pro.3715) and Gene Ontology (https://www.ncbi.nlm.nih.gov/pmc/articles/PMC3037419/, https://pubmed.ncbi.nlm.nih.gov/33290552/) were used to categorize genes. Heatmaps were generated with the heatmap.2() function in gplots.

## HUVEC proliferation, viability and apoptosis assays

To test proliferation, 24, 48, 72, or 96 h post-transfection HUVECs were collected, and cell number was assessed by using a hemocytometer[86]. Viability was further determined by Trypan blue dye exclusion. A flow cytometry based EdU incorporation assay was also performed to assess new DNA synthesis as a proxy for cell proliferation using ClickIT-EdU cell proliferation kit (ThermoFisher, Cat#C10632,). At the indicated times post-transfection, cells were incubated with EdU (10μM) for 3 h before collection and further processed according to the manufacturer's instructions. To measure viability, 48 or 72 h post-

transfection, cells were collected and stained with CCK8 cell viability dye (Cell viability assay kit, Sigma), absorbance recorded at 450 nm. To determine apoptosis, Annexin V apoptosis detection assay kit (ThermoFisher, Cat#88-8005-74) by flow cytometry BD LSRII (BD Biosciences)[88].

## Senescence-associated β-galactosidase staining

We used senescence-associated β-galactosidase staining kit (Cell Signaling Technology, Cat#9860 S). To quantify the staining, we selected four representative microscopic images for each sample. HUVECs were exposed to 100 μM of hydrogen peroxide ($H_2O_2$) for 3 h (positive control). Imaging was conducted on an Invitrogen EVOS microscope equipped with a 10x objective lens. The images were subsequently analyzed and quantified using Fiji/NIH ImageJ software (National Institutes of Health).

## Western blot analysis

At the indicated times, whole-cell lysate isolation and western blot was performed[86,88]. Membranes were probed with the respective primary antibodies (anti-Phospho-AMPKα, AMPKα, Phospho-p44/42 MAPK, p44/42 MAPK, VEGF Receptor 2, Hexokinase II, LDHA, PFKP, PKM2, PDH, ACSL1 (Cell Signaling Technology), (anti-LIPG, Origene), anti-HSP90 (BD Biosciences), anti-CPT1A (Abcam) and anti-CD36 (Proteintech) (all antibody dilution 1: 1000) at 4°C overnight. Membranes were washed with TBS-T followed by incubation with appropriate secondary fluorophore-conjugated antibodies (Invitrogen). Protein bands visualized using the Odyssey Infrared Imaging System (LI-COR Biotechnology), and densitometry performed using ImageJ. Antibody information in Supplementary Table 3.

## HUVEC cord formation assay

After 60 h of ANGPTL4 silencing, HUVECs (70 ×10³ cells/well) were cultured in a 24-well tissue culture plate coated with 250μl of Growth Factor Reduced Matrigel (BD Biosciences)[89].

## HUVEC cell migration assay

A trans-well cell migration assay was performed using the 24-well trans-well chamber system (8 μm pore size, Costar, # 3422)[86]. 60 h post-transfection (75000 cells) were plated onto the inserts (upper side) coated with 0.1% gelatin. The bottom chambers were filled with M199/EBSS medium containing either BSA, VEGF (50 ng/ml) or FBS (20%) and allowed the cells to migrate through the inserts for 6 h, then fixed with 4% PFA followed by staining with eosin (Dip-stain solution II, #VDB-016) and methylene blue solution (Dip-stain solution III, #VDB-016). Cells that were migrated to the outer side of inserts were photographed and total number of cells was counted.

## HUVEC sprouting assay

The sprouting angiogenesis assay was performed in HUVECs upon silencing ANGPTL4[44,86]. 24 h post-transfection, cell media was changed to EGM-2 (Lonza CC-3162). The following day, 1×10⁶ HUVECs were mixed with 2500 beads in EGM-2 medium for 4 h at 37 °C. Coated beads were resuspended in fibrinogen solution (2.0 mg/mL fibrinogen, 0.15 Units/mL of aprotinin) at a concentration of ~200 beads/mL. 0.625 Units/mL of thrombin was added to each well of a 24-well plate and then 0.5 mL of the fibrinogen/bead suspension was added to each well. HUVECs were allowed to undergo morphogenesis for 2-3 days. Beads were then permeabilized in 0.5% Triton-X 100 for 2 h at room temperature, followed by incubation with Alexa Fluor™ 594 Phalloidin (Invitrogen) and DAPI for 3 h at RT. Sprout and branch morphology was analyzed in ImageJ. Sprouts were defined as vessels with at least one nucleus separated from the bead, and branches were defined as vessels with a separate nucleus that originate from an existing sprout.

## Mouse lung endothelial cell isolation and immortalization

MLECs were isolated[86,90], from three pairs of lungs dissected from 3-4 weeks-old wild-type WT or Angptl4^iΔEC mice. Freshly isolated lung tissue was minced with scissors and allowed to digest at 37 °C with 2 mg/mL ≈ 175 u/mg Type I collagenase (Sigma) for 1 h. Lung tissue was further subjected to mechanical disruption by passage ≈ 12 times through a 14-gauge needle and filtration through 70μm steel mesh. ECs were immuno-isolated using sheep anti-rat IgG–coated magnetic beads precomplexed with anti-PECAM-1 antibody (Pharmingen). After 2-3 days, cells were immortalized and re-immunoselected with PECAM-1 magnetic beads. Cells were propagated in EGM-2 media supplemented with EGM-2 microvascular (Lonza).

## Mouse retina vascular system analysis

For retina staining, procedures were followed as we described[86,90]. Briefly, retinas were dissected out and stained for 2 h in a 1:50 solution of Isolectin-B4-647 (Life Technologies), followed by mounting and imaging on a confocal microscope. Retinal area and vascular density were assessed using ImageJ[86].

## Matrigel plug assay

For Matrigel plug angiogenesis assay, male mice (24 weeks old) were subcutaneously implanted with 500 μl of Matrigel containing heparin (0.1 mg/ml), FGF (50 ng/ml), and VEGF (50 ng/ml)[91] (R&D systems, Cat# 3139-FB-025/CF, 493-MV-005/CF). After 7 days, mice were euthanized, and plugs were excised and photographed. Hemoglobin content in the plugs was determined via spectrophotometer[92]. For the analysis of vasculature, Matrigel plugs were fixed in 4% paraformaldehyde for 24 h, and paraffin-embedded and sectioned onto poly L-Lysine coated slides. Slides processed and incubated with Isolectin GS-IB4, Alexa Fluor 647 (Invitrogen) for 2 h at RT and counterstained and mounted with ProLong gold containing DAPI (Invitrogen). Images for micro-vessel structures were taken with a fluorescence microscope (EVOS, Life Technologies).

## Tumor growth, angiogenesis, EdU incorporation and tumor permeability analysis

LLC and B16F10 cells were injected subcutaneously into the dorsal flank of 8- 10-week-old female mice[46,86,90]. Tumor growth was monitored by measuring the length and width of the tumor using a caliper, and tumor volume was determined by: volume = 0.52 × (width)² × (length). All mice were euthanized when tumor reached approximately 1 cm³, and tumors were collected and weighed. Tumors were cut into 5 mm pieces and fixed in 4% PFA on ice for 4 h, followed by overnight incubation in 30% sucrose solution, then embedded in OCT and frozen for immunofluorescence analysis. For EdU incorporation, mice were intravenously injected with 0.1 ml of EdU (10 mg/kg) and 6 h later, mice were euthanized, and tumors were collected and prepared as above indicated. The incorporated EdU was detected by a 'click-It reaction' with Alexa Fluor 647 according to the manufacturer's instructions (ThermoFisher, Cat# C10340), see below for detailed analysis. To determine vascular permeability, mice were intravenously injected with 100μl FITC-Dextran (70 kDa or 4 kDa, 25 mg/mouse)[93]. Extracellular vascular FITC-Dextran fluorescence was analyzed by a fluorescence microscope. Vascular permeability was also assessed by administering Evans Blue (EB) dye as described[94]. An 1% Evans Blue solution saline (100μl/mouse) was injected intravenously. One hour after injection, the mice were euthanized, and tumor tissues were harvested and weighed. The EB dye was then extracted by immersing the tissues in formamide and incubating them at 55 °C overnight. The concentration of EB in the tumor was quantified using spectrophotometric measurements at 620 nm and normalized to the total weight.

## Immunofluorescence staining

Tumors were excised from WT and *Angptl4*[iΔEC] mice and prepared as above indicated. Serial sections of the tumor were cut (5 μm thickness) with a cryostat and mounted on poly L-lysine-coated slides (Thermo-Fisher). Slides were permeabilized with 0.1% Triton X100 in PBS for 10 minutes and blocked in 20% FBS in PBS. For immunostaining, antibodies for CD31 (BD, clone:MEC13.3, Cat# 5502730 was used to detect vessel structures, and anti-Ki67 antibody (R&D) was used to detect proliferating cells. For detecting pericyte coverage, antibodies for NG2 (Invitrogen, clone: 546930, cat# MA5-24247,) and αSMA (Invitrogen, Cat#50-9760-82) were used along with CD31-PE (BD, clone:MEC13.3, Cat #102406). Nuclei were counterstained using ProLong gold containing DAPI (Invitrogen, Carlsbad, CA). For determining EC cycling, EdU and ERG co-staining was performed using anti-ERG antibody (Abcam, Cat# ab92513). Sections were visualized under a fluorescence microscope (EVOS, Life Technology). 3 sections per sample were analyzed and 2-4 images were captured from random areas of each tumor section. Microvessel density was quantified by measuring the CD31[+] vessel-like structures per sample per area. The values per area of individual images were then averaged to obtain the mean value per area-fraction for each tumor. For quantifications ImageJ was used. Antibody information in Supplementary Table 3.

## Single-cell sorting of mouse lung ECs

Tissue homogenates (lung and tumor) were stained with the following antibodies: anti-mouse CD45 (eBioscience, clone 30-F11, #13-0451-82), anti-mouse CD31-PE, as above. For flow cytometry, live cells were gated by staining with LIVE/DEAD™ Fixable Blue Dead Cell Stain Kit (Invitrogen, Cat# L23105,). All antibodies were titrated to determine optimal signal-to-noise separation and minimize background fluorescence. The gating strategy was set up using Fluorescence Minus One (FMO) controls. (Antibody information is in Supplementary Table 3). ECs were identified and sorted as CD45[-]CD31[+] cells using BD FACS Aria (BD Biosciences)[46,88] for RNA isolation to determine *Angptl4* levels via qRT-PCR, as described above. All flow cytometric analyses were performed using a BD LSRII (BD Biosciences).

## Glucose utilization and lactate secretion assay

Cells ($3 \times 10^5$) were cultured in M199/EBSS medium containing 5 mM glucose. After 24 h, medium was replaced with respective medium containing VEGFA or CoCl$_2$ for 24 h, and residual glucose present in the spent medium was monitored using a GOD-POD-based glucose assay kit (Biovision). Consumed glucose was estimated by subtracting the remaining glucose in the medium from the initial concentration in control medium (90 mg/dL). Lactate was measured by using a commercially available lactate assay kit (Biovision) according to the manufacturer's protocol. Values were normalized to total number of cells.

## Oxygen consumption (OCR) and extracellular acidification rate (ECAR) measurements with Seahorse

ECAR and OCR were measured using an XFe96 extracellular flux analyzer (Seahorse, Agilent). Basal ECAR was measured by subtracting the ECAR rate after treatment with 50 mM 2-deoxyglucose (Sigma). Maximum ECAR rate was measured by subtracting the rate after 2-deoxyglucose treatment from the rate after treatment with 1 μM oligomycin A. Basal respiration was determined by subtracting the OCR values after treatment with 1 μM antimycin A (Sigma) and 1 μM rotenone (Sigma). ATP-coupled respiration was measured by treatment with 1 μM oligomycin A (Sigma) and subtracting oligomycin A OCR values from basal respiration.

## Acetyl CoA assay

Acetyl CoA levels in the HUVECs upon silencing ANGPTL4 was determined by commercially available assay kits (Sigma, USA) according to the manufacturer's instructions.

## NAD⁺/NADH assay

NAD/NADH levels in the HUVECs upon silencing ANGPTL4 was determined by commercially available assay kits (Cayman, USA) according to the manufacturer's instructions.

## Enzyme (PFK and HK) activity assay

Enzyme activity was analyzed in cells post transfection and treatments[95]. HUVECs cultured under desired experimental conditions were homogenized in a hypotonic potassium phosphate buffer (20 mM, pH7.5) containing protease inhibitor cocktail (Roche). PFK activity was determined by an enzyme-coupled reaction method. The enzyme reaction was performed in Tris-Cl buffer (50 mM, pH 8.0) containing ATP (0.1 mM), MgCl$_2$ (3.3 mM), NADH (0.1 mM), glyceraldehydes-3-phosphate dehydrogenase (1U/ml), aldolase (1U/ml), triose phosphate isomerase (6 U/ml). The reaction was started by adding 3.3 mM fructose 6-phosphate and absorbance was recorded at 340 nm at every 30 s for 10 min. For analyzing HK activity cell lysate was added to the assay buffer (100 mM KCl, 250 mM MgCl2, 10 mM ATP, 50 mM EDTA, 100 μM glucose, 1U G6PDH and 10 mM NADP). Increase in the absorbance (340 nm) was followed every 30 s for 10 min using a spectrophotometer (SpectraMAX, Molecular Devices) under constant temperature (37 °C). The values were normalized to protein content.

## Uptake of fluorescently labeled glucose and FA

Uptake of glucose and FA in ECs were analyzed by using fluorescently labeled probes: 2NBDG and Bodipy-Fl-C16 respectively according to the manufacturer's instructions. After the indicated incubations conditions, cells were treated with 100 μM of 2NBDG (ThermoFisher, Cat# N13195) or 1 μM of BODIPY-FL-C16 (ThermoFisher,Cat#D3821) for 30 min at 37°C. Cells were washed with PBS, and the uptake of these probes was analyzed by flow cytometry. For in vivo analysis of FA and glucose uptake in ECs, mice were given 2NBDG or BODIPY-FL-C16 either intravenously or orally for 1h[96]. Fluorescence of 2NBDG or BODIPY-FL-C16 in ECs was analyzed either by flow cytometry of single-cell suspensions prepared from different tissues or by the immuno-fluorescence microscopy of the frozen sections of tissues co-stained with anti CD31-PE antibody (BD Bioscience). FACS Gating strategy is in Supplementary Fig. 12.

## Metabolic flux and enrichment studies using MIMOSA (mass isotopomer multiordinate spectral analysis)

MIMOSA analysis was performed to determine the metabolic flux and enrichment of metabolites from glycolysis and the TCA cycle from [U-$^{13}$C$_6$]-glucose[51]. At the indicated time post-transfection, cells were quenched by washing with ice-cold PBS and collected in 150 μl of an ice-cold metabolite extract solution (20% methanol, 0.1% formic acid, 3 mM NaF, 100 μM EDTA, and 1 mM phenylalanine). Metabolite concentrations and $^{13}$C-enrichments were determined by mass spectrometry using a SCIEX 5500 QTRAP equipped with a SelexION for differential mobility separation (DMS). Samples were injected onto a Hypercarb column (3μm particle size, 3×150 mm, ThermoFisher) at a flow rate of 1 mL/min, using a Shimadzu high-performance liquid chromatography (HPLC) system. Metabolites were eluted with a combination of aqueous (A: 15 mM ammonium formate and 10uM EDTA) and organic mobile phase (B: 60% acetonitrile, 35% isopropanol and 15 mM ammonium formate) according to the following gradient: t = 0 min, B = 0%; t = 0.5 min, B = 0%, t = 1 min, B = 40%; t = 1.5 min, B = 40%; t = 2 min, B = 0%; t = 6 min, B = 0%. Metabolite detection was based on multiple reaction monitoring (MRM) in negative mode using the following source parameters: CUR: 30, CAD: high, IS: -1500, TEM: 625, GS1: 50 and GS2: 55. DMS parameters were DT: low, MD: 2-propanol, MDC: low, DMO: 3 and DR: off, while Separation Voltage (SV) and Compensation Voltage (CoV) were optimized individually for each metabolite in order to maximize signal intensity and isobar resolution.

The individual MRM transition pairs ($Q_1/Q_3$) are listed in Supplementary Data 1. Retention times were confirmed with known standards and peaks integrated using El-Maven (Elucidata). The atomic percent excess (APE) was calculated using Polly interface (Elucidata) and corrected for background noise and for natural abundance. All measured metabolites in each sample were normalized by total protein content (Supplementary Data 1).

### Triglyceride lipase and phospholipase A1 activity assay
Total lipase activity in either culture medium or post-heparin plasma was measured by incubating samples with 10% Intralipid/[³H] Triolein emulsion as a substrate[97]. For the analysis of heparin-releasable triglyceride lipase activity in the tissues or in the cells, cells/tissue were homogenized in ice-cold PBS containing 2 mg/ml BSA, 5U/ml heparin, 5 mM EDTA, 0.1%SDS and 1% Triton. Homogenates were centrifuged, and supernatant was separated and 100 μl supernatant was mixed with 10% Intralipid/[³H]-triolein emulsion for 1 h at 25 °C. Triglyceride lipase activity was normalized to tissue weight or protein concentration from cell lysate. Phospholipase A1 activity was determined using a commercially available assay kit (ThermoFisher, Cat#E10219) according to the manufacturer's protocol.

### Fatty acid oxidation assay
FA oxidation in HUVECs was determined using [¹⁴C] palmitate[12]. HUVECs were incubated with reaction mixture (0.5 mmol/L palmitate conjugated to 7% BSA/[¹⁴C]-palmitate at 0.4 mCi/ml) for 30 minutes. The reaction mixture was then transferred to an Eppendorf tube (1.5 ml), the cap of which housed a Whatman filter paper disc pre-soaked with sodium hydroxide. $^{14}[CO_2]$ trapped in the filter paper was then released by using 1 M perchloric acid and moderately agitating the tubes at 37 °C for 1 h. Radioactivity adsorbed onto the filter disc was quantified by a liquid scintillation counter.

### Measurement of superoxide generation
Cellular superoxide ($O_2^{\cdot}$) levels were determined in HUVECs transfected with NS or *ANGPTL4* siRNAs using a fluorescence dye, DHE (ThermoFisher, Cat#D11347), according to the manufacturer's instructions by flow cytometry.

### Preparation of radiolabeled chylomicron
Mice were fasted for 4 h and then injected with the LPL inhibitor poloxamer (1 g/kg). After 1 h, mice were gavaged with an emulsion mixture containing [³H]-triolein (80 μCi) and intralipid 20% emulsion oil in 100 μl final volume/mouse. Plasma was collected 3 h after the gavage and the radiolabeled chylomicron fraction was collected through ultracentrifugation[59].

### Lipid uptake in cells and tissue
Lipid uptake in tissue or in transfected cells was performed by using [³H] triolein[14]. Briefly, for tissue lipid uptake, mice fasted for 4 h were orally administered with 100 μl emulsion containing 2μCi [³H]-triolein. Organs were harvested 2 h after oral gavage and lipids were extracted with isopropyl alcohol-hexane (2:3). For cells in culture, at the indicated time post-transfection they were starved with serum free-culture medium containing fatty acid-free BSA for 3 h followed by incubation with the emulsion above indicated. The lipid layer was separated, and radioactivity (CPM) of [³H]-triolein was analyzed by a liquid scintillation counter.

### Plasma lipid measurements
Blood was collected retro-orbitally from the overnight-fasted (12-16 h) mice, and plasma was separated by centrifugation. HDL-C was separated by precipitation of non−HDL-C, and both HDL-C fractions. Total plasma TAGs and cholesterol were enzymatically analyzed using commercially available kits (Wako Pure Chemicals)[12,14,88].

### Statistics & reproducibility
Statistical analyses were performed using GraphPad Prism. In vitro experiments include at least >2 technical replicates of 3 different biological replicates. All the data are presented as mean ± SEM. Statistical differences measured using two-tailed unpaired nonparametric t test (Mann−Whitney U test). For multiple groups comparisons, two-way ANOVA was applied. Post-hoc analysis using Tukey's multiple comparisons. *P*-values below 0.05 were considered statistically significant. Differentially expressed genes of RNA-seq data were identified as described above in RNA-seq data analysis section. The statistical parameters for each experiment can be found in the figure legends. Study details for reproducibility purposes are detailed in Reporting Summary provided with this paper.

### Reporting summary
Further information on research design is available in the Nature Portfolio Reporting Summary linked to this article.

## Data availability
RNA-seq data that support the findings of this study have been deposited in the Gene Expression Omnibus under accession code GSE211128. Raw steady state metabolite measurement data can be found in Supplementary Data 1. All other data that support the findings of this study are within the article and its Supplementary Information and Source Data files. Source data are provided with this paper.

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

## Acknowledgements

Dr. Christopher Castaldi at Yale Center for Genome Analysis for bulk RNA-seq processing and Rolando Garcia-Milian at Medical library for RNA-seq analysis advice. Chemical Metabolism Core (CMC) for the MIMOSA work. Illustrations were created with BioRender.com. This work was supported at least in part by grants from the National Institutes of Health (R35HL135820 to CF-H; RK is supported by R01 DK127637; KMC is supported F31HL156319 from NIH/NHLBI) and the American Heart Association (AHA 20TPA35490202 to YS).

## Author contributions

B.C., K.M.C., M.S., A.K.S., D.S.U.I., W.D., R.P., R.R. and R.C. performed experiments and analyzed data. R.K., C.F.-H., and Y.S. assisted with experimental design, data analysis and interpretation. Y.S. designed the study. Y.S., B.C. and K.M.C. wrote the manuscript, which was commented on by all authors.

## Competing interests

The authors declare no competing interests.
