## [Peer Review File · Nature Communications]

Suppression of Angiopoietin-like 4 Reprograms Endothelial Cell Metabolism and Inhibits AngiogenesisEditorial Note: Parts of this Peer Review File have been redacted as indicated to maintain the confidentiality of unpublished data.

REVIEWER COMMENTS

Reviewer #1 (Remarks to the Author):

Summary:

The manuscript by Chaube and colleagues studies the role of Angiopoietin-like 4 (ANGPTL4) in endothelial cells (ECs). Using a combination of in vitro and in vivo approaches, they report that Angptl4 deficiency rewires endothelial metabolism and lowers their angiogenic potential. The authors propose that ANGPTL4 mutant ECs adopt a pro- quiescent phenotype as they shift their metabolism towards fatty acid usage. As a result, tumor vascularization is compromised in endothelial-specific Angptl4 knockout mice.

Comments:

The manuscript reports a potentially new function of Angptl4 in the endothelium. However, the data - particularly the in vivo data - is of insufficient depth to fully support the model. One would like to see a more thorough characterization of the EC-specific Angptl4 knockout mice. For instance, what happens when Angptl4 is depleted during physiological vascular expansion? Does loss of Angptl4 impair endothelial parameters affected in the tumor vasculature also in the physiological setting?

The depth of the mechanistic analyses should also be increased. For example, tumor-derived proteolytic fragments of ANGPTL4 have been shown to have both pro- and anti-tumor angiogenic properties (PMID:36269299). Furthermore, a detailed quantification of ANGPTL4 sources seems necessary to pinpoint the specific role of endothelial-derived ANGPTL4. Because tumor cells also secrete ANGPTL4, the use of tumor cell lines deficient for ANGPTL4 (e.g. by shRNA or CRISPR-Cas), would be an insightful control to support the proposed model.

As it stands, the manuscript does not provide compelling evidence for the proposed function of ANGPTL4, and further experimental data is needed to provide mechanistic insight.

Further comments:

1. ANGPTL4 belongs to a family of nine structurally similar proteins. It would be interesting to show the expression levels of ANGPTL4 and the other family members in endothelial cells.

2. Does siRNA-mediated depletion of ANGPTL4 impact endothelial cell death? CCK8 assays are not meant to assay cell viability but rather cell proliferation/growth.

3. What happens to the protein levels of ANGPTL4 (intracellular and serum) in the EC-specific Angptl4 knockout mice?

4. It is challenging to assess several of the authors' claims because of the insufficient resolution of the vascular images. An example is Figure 2C, where the vessel-like structures are difficult to appreciate. Given these limitations, one wonders how accurate the quantifications are. High-quality 3D images would help to substantiate some of the claims.

5. To validate the effects of ANGPTL4 loss on EC proliferation, the authors should perform additional EdU incorporation assays in vitro and in vivo. For the in vivo studies counterstains for endothelial nuclear markers (e.g. ERG) are necessary to unambiguously identify cycling ECs.

6. All the experiments were done in the context of Angptl4 depletion. What happens to endothelial proliferation and metabolism when ANGPTL4 levels are increased?

7. How efficient is the silencing of LIPG in ECs (line 331)? Authors should provide immunoblots and quantifications to support their findings.

Reviewer #2 (Remarks to the Author):

In this manuscript, the authors investigated the role of endogenous ANGPTL4 in endothelial cells and angiogenesis. The main claims are that ANGPTL4 is indispensable to normal angiogenesis, and such a crucial role of ANGPTL4 in angiogenesis was at least partially dependent on its inhibitory effect on endothelial lipase activity and the subsequent shift of metabolic signature from FAO to glycolysis. This study provided solid evidence to show that ANGPTL4 functions in promoting angiogenesis which addressed a controversial question about the role of ANGPTL4 in the field of vascular biology. The data are solid, and novel. I have the following comments to address:

The primary issue to address is the claim that the metabolic changes observed contribute to the angiogenesis phenotype, which is not well defended. The authors convincingly show that loss of ANGPTL4 leads to angiogenic defects, in a manner dependent on inhibition of EL, and that the loss of ANGPTL4 leads to metabolic changes, again in a manner dependent on inhibition of EL. What is not convincingly shown is that the latter leads to the former, i.e. that the angiogenic phenotype is dependent on the metabolic changes. The easiest solution to this is to change the abstract to not make this claim, ie. back off on “Consequently, EL mediated altered FA metabolism reprograms glucose metabolism, promoting a metabolically quiescent phenotype that disfavor angiogenic activation in ECs”, which is not defended by data.

The second major issue to address is that the authors should reconcile their results with the seemingly contradictory results of Carmeliet’s group Schoors et al.; PMID: 25830893 (which is not referenced at all).

Other points:

1. Cell senescence and autophagy should also be ruled out in the anti-angiogenic effects of ANGPTL4 depletion.
2. Can the decrease of KDR, P-ERK also be rescued by EL inhibition (XEN445 and GSK264220A)
3. For permeability assay, Evans blue assay should be included.
4. Is it possible for Lipase/FAO axis to regulate angiogenesis through controlling FOXO1/cMYC pathway?
5. Since EC proliferation also requires carbon contribution from FAO, it is confusing that XEN445 mediated FAO inhibition didn’t show any anti-angiogenic effects by itself. What do the authors think is the difference?
6. Rescue of key phenotypes (e.g. proliferation) should be tested by siLIPG. Specifically, proliferation/angiogenesis should be tested under conditions in Ext. data 5h.
7. Lines 41 (abstract) and 463, 478, 493 in text mention that altered fatty acid metabolism promotes quiescence. This statement is too strong and should be reworded. Ref 42 is referenced, but the statement is not a correct interpretation of the reference: there is no

evidence that FAO promotes quiescence.

minor

8. In line 371, confusing interpretation that “via inhibiting lipase activity”, is it “via increasing...”?

9. In Extended Fig 7c, the representative picture doesn't show consistent outcome with that of quantification on the right.

10. In Fig7e, the representative fig doesn't show consistent outcome with that of quantification on the right.

11. In line 540, the statement “promoting angiogenesis and tumour growth” is opposite to the data Fig2 shown in the manuscript.

12. In line 156, Fig. 2a should be cited instead of Fig. 1a

13. Line 175 contains typo: ANTPL4 should be ANGPTL4

Reviewer #3 (Remarks to the Author):

The paper by Chaube and colleagues is the first to examine the metabolic and angiogenic effects of ANGPTL4 deficiency in endothelial cells (EC). The paper shows that silencing of ANGPTL4 in EC has major effects on a large number of metabolic and angiogenic parameters. The data point toward a huge role of ANGPTL4 in EC metabolism and angiogenesis, which is suggested to be mediated by the inhibitory action of ANGPTL4 on endothelial lipase (EL). Specifically, it is suggested that the inactivation of ANGPTL4 promotes EL-mediated vascular lipolysis, which results in increased fatty acid uptake and utilization in favor of FA oxidation. Although the link between ANGPTL4 and angiogenesis has been extensively studied, the role of ANGPTL4 in EC remains rather unclear, mostly because the published data are very conflicting. Several papers, including the first papers reporting on ANGPTL4 expression in endothelial cells, have found that ANGPTL4 stimulates *in vitro* tube formation and endothelial migration, as well as *in vivo* angiogenesis. By contrast, several other papers report an anti-angiogenic role for ANGPTL4. The inhibitory effect of ANGPTL4 on endothelial lipase has also been previously reported yet the impact of this pathway on EC metabolism and angiogenesis has not been studied. All previous studies that have investigated the role of ANGPTL4 in EC have used recombinant ANGPTL4 protein or an overexpression model. In contrast, Chaube and colleagues have inactivated the ANGPTL4 gene either via *in vitro* siRNA-based silencing or via *in vivo* Cre—mediated gene inactivation.

The main novelty of the paper is the finding that ANGPTL4 silencing appears to have a huge effect on glycolytic and fatty acid oxidative pathways, which is associated with major changes in the expression of numerous metabolic genes. Furthermore, it is shown that endothelial ANGPTL4 deficiency *in vivo* has anti-tumor and anti-angiogenic effects, which is explained by increased lipase activity, suggesting that local lipase activity in EC is important for the regulation of endothelial functions and tumor angiogenesis.

The paper has clear merit, as indicated above. My main concerns relate to the mechanistic framework for the paper. To my understanding, there is no precedent and no biochemical basis for EL regulating metabolic pathways in EC in the absence of external phospholipids or triglycerides as substrate. For example, Nakajima found that EL upregulates the expression of fatty acid oxidation-related enzymes and intracellular adenosine triphosphate

accumulation, but these studies were done in the presence of high-density lipoprotein, which the main EL substrate (PMID: 23460280). Most of the experiments shown were conducted in EC in the absence of an EL substrate. The data by Chaube imply that the function of EL goes beyond being a lipase, for instance by showing that ANGPTL4 regulates the uptake of non-esterified fatty acids by EC via EL, a process that doesn't require a lipase enzyme. It is not impossible that the function of EL goes beyond that of a lipase but this needs to be shown in the paper to validate the suggested sequence of events occurring upon ANGPTL4 silencing.

Major comments

1) What is the evidence that the observed changes in fatty acid and glucose metabolism upon ANGPTL4 silencing in EC account for the effects of ANGPTL4 silencing on angiogenic functions? Couldn't it be the other way around? Currently, the temporal relationship between the changes in EC function and metabolism is unclear. Based on the data presented, it cannot be concluded that metabolic rewiring precedes angiogenic changes, such as EC migration, tube formation etc. The conclusions of the paper should be redacted accordingly.

2) Previously, the C-terminal portion of ANGPTL4 was shown to bind to specific integrins, which likely accounts for several of the non-metabolic functions of ANGPTL4. Concerning angiogenesis, it was shown that ANGPTL4 counteracts hypoxia-driven vascular permeability through integrin $\alpha v \beta 3$ binding, modulation of VEGFR2-*Src* kinase signaling, and endothelial junction stabilization (PMID: 27577973). Another paper found that the C-terminal portion of ANGPTL4 binds and activates integrin $\alpha 5 \beta 1$ -mediated *Rac1*/*PAK* signaling, ultimately leading to endothelial disruption (PMID: 21841165). Do the authors have specific data that rule out the role of integrin signaling in mediating the observed effect of ANGPTL4 silencing on angiogenic functions?

3) The experiments in figure 5 that show increased lipase activity and fatty acid uptake in EC treated with ANGPTL4 siRNA were done in the presence of exogenous sources of TG and fatty acids, respectively. No exogenous sources of TG and fatty acids were seemingly provided in the ANGPTL4 silencing experiments shown in the previous figures. Accordingly, it is difficult to envision how, in the absence of any EL substrate, increased lipase activity and fatty acid uptake can account for the observed metabolic and angiogenic effect upon ANGPTL4 silencing.

4) EL is a lipase enzyme and is not known to be directly involved in the uptake of free fatty acids by cells. Hence, how can the increased uptake of fluorescently-labeled fatty acids upon ANGPTL4 silencing be attributed to EL? The experiments with XEN445 are very useful and convincing but also raise major questions. How can an EL inhibitor reduce the uptake of a fluorescently-labeled fatty acid? How can an EL inhibitor regulate the expression of numerous metabolic genes (Fig 6c)? There is no mechanistic basis for EL impacting the cellular uptake of non-esterified fatty acids. The paper would be much more persuasive if data could be provided showing that overexpression or silencing of LIPG (without modulating ANGPTL4) influences the key pathways affected by ANGPTL4 silencing in figures 1,2 and 4.

5) A crucial experiment to do is to silence LIPG in EC and determine whether the key effects of ANGPTL4 siRNA described in figures 1,2 and 4 (especially RNAseq analysis) can be abolished by EL silencing. This experiment can reveal whether EL mediates the metabolic effects of ANGPTL4 in EC. The present manuscript only shows the effects of silencing of LIPG on lipid accumulation in EC treated with ANGPTL4 siRNA (Extended Data Fig. 5h). It is important to comprehensively study the effects of co-silencing of ANGPTL4 and LIPG.

6) Another important experiment to do is to study the effect of exogenous ANGPTL4. Specifically, in a new experiment, EC treated with ANGPTL4 siRNA should be co-treated with exogenous ANGPTL4 (or conditional medium of ANGPTL4-overexpressing cells vs control medium) to see if the effects of ANGPTL4 silencing on metabolic pathways, cell migration/adhesion, and angiogenesis can be abolished by external ANGPTL4. This experiment could be conducted with N-terminal ANGPTL4 and C-terminal ANGPTL4 to determine which part of the ANGPTL4 protein mediates the observed effects.

7) Since this paper is the first to report on the phenotype of EC-specific ANGPTL4 deficient mice, it would be of interest to have information on basic metabolic parameters, including plasma TAG, cholesterol, non-esterified fatty acids, and glucose. Having plasma lipoprotein profiles would be even better.

8) The link between EC-derived ANGPTL4 and metabolic pathways is entirely based on studies in HUVEC using siRNA. The manuscript would greatly benefit from a comparative analysis of the gene expression profiles (RNAseq) of primary EC isolated from WT and *Angptl4iΔEC* mice, confirming the key metabolic outcomes of ANGPTL4 silencing observed in HUVEC cells, including the stimulatory effect of ANGPTL4 deficiency on fatty acid oxidation genes and inhibitory effect on glycolysis genes.

9) The paper builds on evidence that ANGPTL4 is an EL inhibitor. However, this evidence is weak. Plasma HDL levels are not altered by ANGPTL4 deletion in mice or the E40K variant in humans consistent with reduced EL inhibition (plasma HDL levels are increased in E40K carriers rather than decreased). Also, a recent genetic mimicry analysis does not support EL inhibition by ANGPTL4 in humans, although the data do support EL inhibition by ANGPTL3/ANGPTL8 (PMID: 36372100). The paper should put more effort into investigating the potential inhibition of EL by ANGPTL4, using the EC-specific ANGPTL4-deficient mouse model or other models.

Minor comments

1) The paper would benefit from experiments in HUVEC cells cultured under hypoxic conditions, as opposed to using CoCl₂ to induce chemical hypoxia.

2) The manuscript would benefit from data showing that there is a decrease in ANGPTL4 protein in the Cd45⁻ and CD31⁺ cells isolated from the lung of *ANGPTL4iΔEC* mice.

3) The following question may be better experimentally studied in a future publication but perhaps the authors can already speculate in the discussion. How do the authors envision that increased fatty acid uptake following enhanced EL activity leads to the massive changes in gene expression observed in EC treated with ANGPTL4 siRNA? What transcription factor could for example mediate the major effects on glycolytic gene expression?

4) There is a problem with the resolution of the bottom heatmap of figure 3c. Same for figure 1e.

5) In the methods, please indicate which ANGPTL4 exon(s) is amplified by the qPCR that is used to validate the effective knock-down of ANGPTL4 in the Cd45⁻ and CD31⁺ cells isolated from the lung of *Angptl4iΔEC* mice.

6) Line 242, 255: shouldn't it be anaerobic glycolysis?

7) It is incorrect to say that EC-derived ANGPTL4 is essential for EC lipolysis and FA uptake (lines 305-307). Rather, EC-derived ANGPTL4 may inhibit EC lipolysis and fatty acid uptake. Please adjust.

Point by point response for original submission (NCOMMS-22-49560-T).

Point by point response to all of the comments raised the Reviewers.
Reviewer's comments are bold and italicized and followed by our answers.

Reviewer #1:

Summary:

The manuscript by Chaube and colleagues studies the role of Angiopoietin-like 4 (ANGPTL4) in endothelial cells (ECs). Using a combination of in vitro and in vivo approaches, they report that Angptl4 deficiency rewires endothelial metabolism and lowers their angiogenic potential. The authors propose that ANGPTL4 mutant ECs adopt a pro-quiescent phenotype as they shift their metabolism towards fatty acid usage. As a result, tumor vascularization is compromised in endothelial-specific Angptl4 knockout mice.

We sincerely appreciate the Reviewer's comments, especially the recognition that our manuscript potentially reports a new function for ANGPTL4 in the endothelium. As described in the Reviewer's comments, that relates to the regulation of EC metabolism and angiogenic responses. We are confident that the additional data included in this revised version, which includes the characterization of *Angptl4* in physiological vascular expansion, will respond satisfactorily to the concerns raised, while improving the overall strength of our manuscript. The Reviewer's comments are italicized, with our responses provided immediately after.

Comments:

The manuscript reports a potentially new function of Angptl4 in the endothelium. However, the data - particularly the in vivo data - is of insufficient depth to fully support the model. One would like to see a more thorough characterization of the EC-specific Angptl4 knockout mice. For instance, what happens when Angptl4 is depleted during physiological vascular expansion? Does loss of Angptl4 impair endothelial parameters affected in the tumor vasculature also in the physiological setting?

The depth of the mechanistic analyses should also be increased. For example, tumor-derived proteolytic fragments of ANGPTL4 have been shown to have both pro- and anti-tumor angiogenic properties (PMID:36269299). Furthermore, a detailed quantification of ANGPTL4 sources seems necessary to pinpoint the specific role of endothelial-derived ANGPTL4. Because tumor cells also secrete ANGPTL4, the use of tumor cell lines deficient for ANGPTL4 (e.g. by shRNA or CRISPR-Cas), would be an insightful control to support the proposed model.

As it stands, the manuscript does not provide compelling evidence for the proposed function of ANGPTL4, and further experimental data is needed to provide mechanistic insight.

We acknowledge the Reviewer's concern regarding the depth of the *in vivo* characterization of the EC-specific *Angptl4* knockout mice. Our study was mostly focused on adult pathological tumor angiogenesis and the Reviewer wonders about the role of endothelial *Angptl4* during physiological vascular expansion. Previous work by Dr. Germain's group (PMID21832056) showed decreased perinatal retinal angiogenesis when *Angptl4* was deleted in a global manner. However, the participation of endothelial *Angptl4* was not specifically addressed. As suggested, we evaluated whether the absence of *Angptl4* in EC affects physiological retinal angiogenesis. Briefly, we found that endothelial KO of *Angptl4* in neonatal mice did not affect retinal vascular expansion as assessed at postnatal day 7. We also confirmed that EC *Angptl4* KO in postnatal mice does not affect completion of the formation/maturation of the superficial plexus by analyzing retinas at postnatal day 14 (new **Supplementary Fig. 2**). These results suggest that endothelial *Angptl4* has minimal effects on the development of healthy vasculatures, while promoting pathological angiogenesis (as we showed in the original and revised version of the manuscript). This also suggests that the absence of

additional sources of *Angptl4* could contribute to the overall phenotype observed by the authors (PMID21832056).

In line with this, the Reviewer wisely pointed in reference to a publication (PMID36269299) that went online right after we submitted our manuscript for review to *Nature Communications*: that different sources of ANGPTL4 (e.g., from tumor cells) could provide confounding effects since different tumor-derived forms of ANGPTL4 have been described to show both pro- and anti-tumor angiogenic properties. To increase the depth of the mechanistic analyses and the participation of endothelial *Angptl4* to tumor-induced angiogenesis, as per the reviewer's suggestion, we performed additional tumor angiogenesis experiments. As indicated, we stably knocked down (KD) the expression of *Angptl4*, using a short hairpin approach, in the tumor cell lines that we used in the initial submission of our manuscript (i.e., LLC and B16F10). The expression of *Angptl4* was barely detectable in both newly generated stable cell lines, referred to as LLC^{*Angptl4-KD*} and B16 F10^{*Angptl4-KD*} (see **Rev #1 Fig. 1A**). However, while the two newly generated LLC^{*Angptl4-KD*} had a proliferative response similar to the parental LLC (expressing *Angptl4*), both of B16F10^{*Angptl4-KD*} cells exhibited increased proliferative response when compared to control B16F10 cells (see **Rev #1 Fig. 1B**). This suggested a differential role for ANGPTL4 in controlling cancer cell proliferative response, which agrees with previous reports. To avoid adding another variable and potential engraftment differences, we chose LLC^{*Angptl4-KD*} cells (short hairpin 2) to determine the contribution of endothelial *Angptl4* to tumor angiogenesis by implanting them into WT and *Angptl4*^{*iΔEC*} mice. Briefly, we found that *Angptl4*^{*iΔEC*} mice also developed smaller tumors compared to WT. This effect was accompanied with a reduction of CD31⁺ vessel-like structures. These new results reinforce the role of endothelial ANGPTL4 in regulating pathological tumor angiogenesis, since in the absence of cancer cell-derived ANGPTL4, *Angptl4*^{*iΔEC*} mice still developed smaller tumors compared to WT. These new results as new **Supplementary Fig. 4** are described and discussed accordingly within the manuscript.

Although physiological and pathological angiogenesis share many similarities, some distinctions between these two events are overt, implying subtle differences in underlying mechanisms. To this regard, the interplay of EC *Angptl4* with the described regulators of angiogenesis in physiological settings vs pathological settings could be different, an area that warrants future investigation. We believe that these new experiments are relevant for the involvement endothelial ANGPTL4 in tumor angiogenesis, so we have included them as new **Supplementary Fig. 4**. All these new results are described and discussed accordingly within the manuscript.

Rev #1, Fig. 1: Differential Impact of ANGPTL4 depletion on cell proliferation in mouse cancer cells (A) Quantitative real-time PCR (qRT-PCR) demonstrates the reduced expression levels of *Angptl4* in LLC and B16F10 cells, following stable knockdown achieved through lentiviral-mediated shRNA transduction. (B) Cell count analyses for both LLC and B16F10 stable clones (EV vs. sh*Angptl4*) reveal varying effects on cell proliferation rates.

Further comments:

1. ANGPTL4 belongs to a family of nine structurally similar proteins. It would be interesting to show the expression levels of ANGPTL4 and the other family members in endothelial cells.

As suggested by the reviewer, we have checked the gene expression of all of the ANGPTL family proteins in both HUVECs and mouse-derived lung endothelial cells (MLECs). In agreement with previous reports that indicated specific expression of ANGPTL3 in the liver (PMID10644446), we did not detect any

transcripts of *ANGPTL3* in HUVECs. Similar data were found for *ANGPTL8*, the most atypical member in the ANGPTL family that shares structural homology with the N-terminal domains of *ANGPTL3* and *ANGPTL4*, as well as functional similarity (LPL binding) with them (PMC5913278). As for the remaining members, we only found *ANGPTL4* and 2 abundantly expressed in HUVECs. Importantly, KD of *ANGPTL4* did not affect *ANGPTL2* levels. As indicated above we also analyzed the expression of all ANGPTL family member in MLECs isolated from WT and *Angptl4^{ΔEC}* mice to provide additional characterization of our mouse model. We found that WT-MLECs express *Angptl1*, 2, 4 and 7 but not the other members. Importantly, we did not see any compensatory increase in the expression of any other ANGPTLs in *Angptl4* KO MLECs. We have included these results as new **Supplementary Fig. 1a** and **Supplementary Fig. 3b** and described and discussed them accordingly.

2. Does siRNA-mediated depletion of ANGPTL4 impact endothelial cell death? CCK8 assays are not meant to assay cell viability but rather cell proliferation/growth.

We agree with the Reviewer that by only using CCK8 it is difficult to unequivocally assess the effects on cell death. CCK8 allows a sensitive colorimetric assay for the determination of the number of viable cells in both proliferation and cytotoxicity assays. As suggested, we performed a more direct cell viability assay by performing Propidium Iodide (PI) and Annexin V staining via flow cytometry. We did not find that the KD of *ANGPTL4* in ECs reduced cell viability, since the percentage of non-viable cells (PI⁺/AnnexinV⁺ plus AnnexinV⁺ cells) was similar to that observed in the control condition. We have included this result as new **Supplementary Fig. 1d** and described and discussed them accordingly.

3. What happens to the protein levels of ANGPTL4 (intracellular and serum) in the EC-specific Angptl4 knockout mice?

We agree with the Reviewer that it would be very interesting to know the protein levels of *ANGPTL4* in *Angptl4^{ΔEC}* mice. However, one of the major limitations of our study is the unavailability of a reliable and specific antibody to detect mouse *ANGPTL4*. We have tried several commercially available antibodies without success. We have accordingly discussed this limitation in the discussion section of the revised manuscript.

4. It is challenging to assess several of the authors' claims because of the insufficient resolution of the vascular images. An example is Figure 2C. where the vessel-like structures are difficult to appreciate. Given these limitations, one wonders how accurate the quantifications are. High-quality 3D images would help to substantiate some of the claims.

We apologize for the low-resolution images that were included, to reduce the overall size of the figures, in the original submission of our manuscript. We have now replaced them with pertinent high-resolution images. We are confident that they reflect the quantification that was initially performed with these images. We agree with the reviewer that the quantification of “vessel-like structures” from the H&E provided in the original Fig. 2C is difficult to interpret. The cross-section of the plugs stained with H&E aimed to show the extent of the acellular area, which can be attributed to the decrease of vascular cells as shown more specifically in Fig. 2d (Now **Fig. 2c** in the revised manuscript) using isolectin B4 staining. Since the H&E does not add further information other than what we have already described with the hemoglobin and isolectin B4 quantification, we opted to remove it from the manuscript.

5. To validate the effects of ANGPTL4 loss on EC proliferation, the authors should perform additional EdU incorporation assays in vitro and in vivo. For the in vivo studies counterstains for endothelial nuclear markers (e.g. ERG) are necessary to unambiguously identify cycling ECs.

In the original submission of our manuscript, we performed a BrdU incorporation assay *in vitro* (original **Supplementary Fig. 1b**) to test the effect on endothelial proliferation when *ANGPTL4* was silenced. The EdU incorporation assay is similar to the BrdU incorporation assay, as both evaluate DNA synthesis and hence cell proliferation. However, the latter needs to denature nuclear DNA and then use a specific detection antibody for BrdU. As suggested, we performed additional EdU incorporation *in vitro*. As expected, and in line

with the previous BrdU assay, *ANGPTL4* KD reduced EdU incorporation when compared the NS control conditions (see new **Supplementary Fig. 1b**, previous BrdU results have been replaced).

The detection of EC proliferation *in vivo* was performed by immunofluorescence co-staining of Ki67 (nuclear antigen marker of active cell proliferation in the normal and tumor cell populations) and CD31 (EC surface marker). As advised, immunostaining of ERG (a highly EC specific transcription factor) was also performed, to unambiguously show the cycling ECs *in vivo*. We repeated the LLC isograft tumor model in *Angptl4^{iΔEC}* mice and performed EdU incorporation together with ERG immunodetection. We are thankful for this suggestion since it reinforces our previous results. Indeed, we also observed reduced intratumor-endothelial proliferation in *Angptl4^{iΔEC}* mice when compared to WT mice (see new **Fig. 2g**). In the revised version of the manuscript, we have included these results as well as those requested by the Reviewer (see **Fig. 2g** and **Supplementary Fig. 1b**) and described them accordingly.

6. All the experiments were done in the context of *Angptl4* depletion. What happens to endothelial proliferation and metabolism when *ANGPTL4* levels are increased?

This is an interesting point raised by the Reviewer. As could be appreciated within the manuscript, we also wondered about the role of *ANGPTL4* when its levels were increased. Interestingly, *ANGPTL4* is one of the most strongly upregulated genes when ECs are subjected to hypoxia (PMC6593883). Furthermore, hypoxia-induced transcriptional and functional changes are associated with increased angiogenic responses and increased metabolic glycolytic activity. Thus, we hypothesized that hypoxia-mediated induction of *ANGPTL4* helps to maintain a pro-angiogenic and glycolytic phenotype since in its absence converse effects are observed. Therefore, instead of performing overexpression experiments (with their own limitations), we opted to assess the pathophysiological function of increased EC *ANGPTL4*. As described, we performed RNA-seq analysis and Seahorse experiments in HUVECs upon silencing *ANGPTL4* in the absence or presence of the hypoxia-mimetic CoCl_2 (See **Fig. 3c** and **Fig. 4f**). Briefly, increasing the levels of *ANGPTL4* via treating HUVECs with CoCl_2 led to increased glycolytic and angiogenic gene expression programs and enhanced glycolytic metabolism which is agreement with previous reports. Interestingly, silencing *ANGPTL4* prevented the hypoxia-mediated regulation of metabolic gene expression, reduced glycolysis, as indicated by reduced extracellular acidification rate (ECAR), under basal and stimulated conditions. This suggests that the angiogenic induction of *ANGPTL4* helps to maintain lipase activity that is otherwise reduced to diminish FA uptake and oxidation in favor of glycolysis. More detailed description and discussion of these results could be found within the manuscript. These results suggest that *ANGPTL4* regulates both basal and angiogenic stimulated metabolic transcriptional programs in ECs that contribute to angiogenic responses.

7. How efficient is the silencing of *LIPG* in ECs (line 331)? Authors should provide immunoblots and quantifications to support their findings.

As indicated, we have now included the *LIPG* immunoblot and quantification to further confirm the efficiency of *LIPG* silencing in ECs. Also, we quantify the level of secreted EL in the heparinized culture medium upon silencing its expression. These new data are now included in **Supplementary Fig. 7c** and **d** and described in the revised version of the manuscript.

Reviewer #2:

In this manuscript, the authors investigated the role of endogenous ANGPTL4 in endothelial cells and angiogenesis. The main claims are that ANGPTL4 is indispensable to normal angiogenesis, and such a crucial role of ANGPTL4 in angiogenesis was at least partially dependent on its inhibitory effect on endothelial lipase activity and the subsequent shift of metabolic signature from FAO to glycolysis. This study provided solid evidence to show that ANGPTL4 functions in promoting angiogenesis which addressed a controversial question about the role of ANGPTL4 in the field of vascular biology. The data are solid, and novel. I have the following comments to address:

We are grateful to the Reviewer for acknowledging that our study “provided solid evidence to show that ANGPTL4 functions in promoting angiogenesis which addressed a controversial question about the role of ANGPTL4 in the field of vascular biology.” The Reviewer also noted some issues and other points (see below) to be addressed and clarified. Despite this, the Reviewer considered that “the data are solid, and novel.” Firstly, we want to sincerely thank the reviewer for recognizing the importance and novelty of our manuscript. We are grateful for all your insightful comments and recommendations. We believe that in responding to all of them, we have substantially improved our manuscript. Reviewer’s comments are italicized and followed by our answers.

The primary issue to address is the claim that the metabolic changes observed contribute to the angiogenesis phenotype, which is not well defended. The authors convincingly show that loss of ANGPTL4 leads to angiogenic defects, in a manner dependent on inhibition of EL, and that the loss of ANGPTL4 leads to metabolic changes, again in a manner dependent on inhibition of EL. What is not convincingly shown is that the latter leads to the former, i.e. that the angiogenic phenotype is dependent on the metabolic changes. The easiest solution to this is to change the abstract to not make this claim, ie. back off on “Consequently, EL mediated altered FA metabolism reprograms glucose metabolism, promoting a metabolically quiescent phenotype that disfavor angiogenic activation in ECs”, which is not defended by data.

Based on the Reviewer’s comment, we realized that our summary of the results is not fully convincing since we cannot claim that the “angiogenic phenotype is dependent on the metabolic changes”, which is not “well-defended”. We fully agree with the Reviewer on that point, and we apologize for the improper wording of manuscript’s abstract with this misleading conclusion. It is indeed a “chicken-and-egg” situation difficult to solve. As suggested, we have accordingly revised this conclusion in the abstract. We have additionally revised the whole manuscript to avoid any further ambiguities. For instance, we make sure that we did not claim that the angiogenic or metabolic effects mediated by the absence of ANGPTL4 are “in a manner dependent on...”, since we think is more appropriate to reflect that the effects are mediated “at least in part” or mediated “to a great extent” since this is supported by our data. See also our response to other points #2.

The second major issue to address is that the authors should reconcile their results with the seemingly contradictory results of Carmeliet’s group Schoors et al.; PMID: 25830893 (which is not referenced at all).

We apologize for this clear oversight from our end. We believe that our results are not fully contradictory to those in the report above indicated. The report of Schoors S et al. showed the importance of FAO in the regulation of angiogenic responses (vessel sprouting) of ECs by KD CPT1A (rate-limiting enzyme of FAO) both *in vitro* and *in vivo*. Absence of endothelial CPT1a caused vascular sprouting defects due to reduced proliferation without affecting migratory responses. Furthermore, inhibition of FAO, using etomoxir, led to similar results, while overexpression of CPT1a produced the converse effects (PMID25830893). The more obvious difference is that in our setting, ECs show “forced” increased of FAO due to an enhanced FA uptake produced by increased vascular lipolysis. Increased FA uptake and oxidation is counter-regulated, diminishing the rate of glycolysis, which can also alter both proliferative and migratory responses of EC, in agreement with a previous report from the same group (PMID23911327). However, the report of Schoors et

al. is related to basal FAO-derived carbons for angiogenesis. In their setting, there were no changes in FA uptake, while glycolysis was not altered (PMID25830893). However, in our case, the absence of *ANGPTL4* promotes endothelial lipase (EL)-mediated vascular lipolysis, which results in an enforced increase of FA uptake. As such, ECs encounter an increased availability of FAs. In this condition, there is a switch in fuel preference (glycolysis vs. FA β -oxidation). Indeed, our metabolic flux analyses showed a preference for FA utilization in *ANGPTL4* KD ECs, which suppressed their ability to metabolize glucose efficiently. In line with this, we also observed increased relative and absolute rates of FAO. However, as shown by Schoors et al, CPT1a KD ECs did not utilize FA for FAO, and glucose utilization was not altered since glycolytic activity remained equal to that of control conditions. Furthermore, FA levels were not altered. In our case, increased FA uptake/oxidation generated acetyl-CoA and citrate. Increased citrate inhibited PFK activity which is in line with previous findings (PMID24445237). However, in CPT1a KD ECs the levels of citrate were reduced. In our settings, even though FAO was increased, we observed hampered proper glucose utilization for angiogenic responses. This is in line with the report that showed that ECs heavily rely on glycolysis for energy sources and vessel growth, and while FAO contributes to ATP generation to a lesser extent, its main role is to provide carbons for *de novo* nucleotide synthesis for DNA replication (PMID25830893). We do not think that nucleotide synthesis should be altered in our case. In our case, inhibition of the increased FAO by etomoxir reduced the enforced utilization of FA, thus mediating a partial rescue of the impaired glycolytic flux. Furthermore, restoration of the inhibition of EL (lost after *ANGPTL4* KD) with XEN 445, partly reverses the metabolic switch and normalizes glycolytic metabolism (see also our response to other points #5). Thus, the effects on EC angiogenic responses are mostly related to the inefficient glucose utilization and metabolism. Therefore, induction of *ANGPTL4* expression in endothelial cells is likely aimed to ensure inhibition of lipase activity thus avoiding EC lipolysis and increased FA uptake that may hamper proper glucose utilization for angiogenic responses. As suggested, we have briefly discussed the above indicated paper by Carmeliet's group (PMID25830893) in relation to our present results in the context of *ANGPTL4* KD in ECs.

Other points:

1. Cell senescence and autophagy should also be ruled out in the anti-angiogenic effects of *ANGPTL4* depletion.

Since after *ANGPTL4* KD we found decreased proliferation and gene expression signatures that may be linked to senescence, it is plausible that some of the anti-angiogenic effects mediated by *ANGPTL4* KD could be related to an induced senescent phenotype. It is important to note that senescence biomarkers are not necessarily specific to senescent cells, as some markers are shared by quiescent or even apoptotic cells as well. However, we did not find induced apoptosis after *ANGPTL4*. Thus, we performed senescence associated β -Galactosidase (SA- β -gal) staining, which is a widely adopted biomarker of cellular senescence to rule out whether if reduced proliferation is associated to an irreversible cell cycle arrest observed in senescence or to the acquisition of quiescent phenotype. As shown in the figure (**Supplementary Fig. 1e**), silencing *ANGPTL4* did not induce presence of SA- β -Gal positive cells, as an indication of SA- β -gal activity, when compared to control cells or to the effect produced by a positive control for senescence. These new data have been included in the revised manuscript.

Autophagy has been implicated in the regulation of pathological angiogenesis (PMC6460396). Additionally, it has been described to play an important role in the homeostasis of lipid metabolism and redox balance which is required for maintaining EC functions (PMID: 21127245, PMID: 19339967). Thus, autophagy might be important for maintenance of these homeostatic processes. As per the Reviewer's request, we performed an in-depth analysis of our RNA seq data focusing on the autophagy-associated genes. Additionally, we measured endogenous LC3 and p62 levels through immunoblotting, to determine whether autophagy is involved in the phenotype observed after *ANGPTL4* KD. Pathway analysis did not show autophagy as a significantly regulated pathway; indeed, genes implicated in autophagy were both upregulated and downregulated (**Rev #2 Fig. 1A**). Despite this, we checked LC3 conversion (LC3-I to LC3-II) and degradation (diminishment) of LC3-II to monitor autophagy, together with the detection of p62, since it is itself degraded by autophagy (PMID25484342). Briefly, we found increased levels of p62 in *ANGPTL4* KD ECs (**Rev #2 Fig. 1B**), suggesting decreased autophagy. The ratio of LC3-II to LC3-I was reduced, which could

suggest an increase in autophagosome formation, leading to enhanced autophagic degradation. On the contrary, this could also be the result of decreased autophagic turnover, resulting in a buildup of LC3-II labeled autophagosomes. However, the decreased ratio was not due to increased LC3-II levels since they were unchanged; rather it is due to increased LC3-I levels (Rev #2 Fig. 1B), which could indicate either unaltered or defective autophagy (PMID25484342). Although not entirely conclusive, these results may suggest a reduction in autophagy following ANGPTL4 knockdown (KD). Previous studies have shown that the induction of autophagy is associated with decreased endothelial cell (EC) angiogenic functions (PMID: 29330051) and reduced VEGFR2 levels (PMC: 3463541). However, the overall role of autophagy in the regulation of angiogenesis could be context-dependent (PMC: 6225658). Therefore, our current results do not necessarily indicate that autophagy plays a role in the reduced angiogenic functions observed in ANGPTL4 KD ECs. We believe that more detailed autophagy flux analyses are needed to accurately determine the impact of ANGPTL4 silencing on autophagy and its potential link to observed changes in angiogenic functions. On a related note, increased fatty acid (FA) uptake and oxidation could potentially alter the cellular redox environment, leading to suppressed angiogenesis. Elevated levels of autophagy may serve as a compensatory mechanism to balance this altered redox state (PMID: 30692642, PMID: 26780888). In ECs with ANGPTL4 knockdown, we did observe enhanced FA uptake and oxidation. However, intriguingly, we did not find evidence of oxidative/ER stress or cell death, as illustrated in Rev #2 Fig. 1C, Supplementary Fig. 6I, and Supplementary Fig. 1d. Given the current data, we cannot definitively conclude that autophagy is involved in the phenotype observed post-ANGPTL4 KD. Therefore, we have chosen not to include these findings in the revised manuscript, unless the reviewer suggests otherwise.

Rev #2, Fig. 1: ANGPTL4 KD in ECs Does Not Appear to Induce Autophagy.

A) Heatmaps display transcript levels of autophagy pathway genes in both ANGPTL4 knockdown (KD) and non-silenced (NS) conditions. The left panel reveals the downregulation of autophagy-related genes following ANGPTL4 KD, while the right panel shows genes that are upregulated in the same condition. B) An immunoblot illustrates the protein levels of p62 and both LC3-I and LC3-II. The right panel provides a quantification of LC3-I, LC3-II, and the ratio of LC3-II to LC3-I. C) Cellular oxidative stress levels were assessed by staining with the oxidative stress (H₂O₂) probe, CellROX.

2. Can the decrease of KDR, P-ERK also be rescued by EL inhibition (XEN445 and GSK264220A)

This is an interesting point raised by the Reviewer. As initially described in our manuscript, the experiment performed in the presence of XEN445 aimed to further confirm the involvement ANGPTL4-mediated inhibition of EL in the regulation of EC metabolism and angiogenic responses, a mechanism that has not been considered/described in the context of the role of ANGPTL4 in EC biology. As described in our manuscript, EL inhibition was able to efficiently rescue the effect on key metabolic pathways such as glycolysis and fatty acid metabolism. Regarding the VEGF signaling pathway, and as acknowledged in the original version of our manuscript, these genes were not fully rescued via XEN445 treatment or EL silencing. A partial rescue was observed for the expression of some genes involved in angiogenesis regulation (**Fig. 6c**). Further in-depth analysis of our RNA-seq performed in HUVECs upon silencing of *ANGPTL4* in the absence or presence of XEN445 (see **Fig. 6**) suggests that EL inhibition upon *ANGPTL4* KD is not able to rescue the *ANGPTL4* KD-mediated reduced expression of *KDR* (see **Rev #2 Fig. 2**). We also performed WB analysis and found that lipase inhibition is not able to restore the decreased protein level of VEGFR2 upon silencing *ANGPTL4* (**Rev #2 Fig. 2**). However, and in agreement with previous reports and as discussed in our manuscript, other mechanisms not directly related to its action on EL might additionally participate in the regulation of EC angiogenic responses which is in line with the above-described results (see also our response to other point # 1). We have included an additional experiment with co-silencing of EL to further corroborate this lack of complete rescue of *ANGPTL4* KD on angiogenic functions (see response to Reviewer 3). In the original submission, we already acknowledge that EL inhibition partially rescues some of the above-indicated processes, as we described above, and we have carefully revised the manuscript to avoid any misleading conclusion. In any case, we believe that the lack of complete rescue of some of the indicated processes does not override the involvement of ANGPTL4-mediated action on EL in the regulation of EC metabolism and angiogenic responses. Since the lack of rescue of *KDR* protein levels may explain the inefficient rescue of the angiogenic responses, this effect requires further investigation to evaluate EL-independent mechanisms that are out of the scope of our present finding. Thus, we opted not to include these results in the revised version of the manuscript as they do not change the conclusions of our main findings. However, if the reviewer considers otherwise, we are prepared to include them.

Rev #2, Fig. 2: Inhibition of EL Fails to Reverse Reduced KDR (VEGFR2) mRNA Levels Induced by ANGPTL4 KD. (Left Panel) The panel shows the normalized log₂ expression levels of KDR (also known as VEGFR2) in HUVECs after ANGPTL4 knockdown. The expression levels are presented both with and without treatment using the lipase inhibitor XEN445. (Right Panel) An immunoblot analysis displaying the VEGFR2 protein levels in HUVECs post-ANGPTL4 knockdown.

3. For permeability assay, Evans blue assay should be included.

To further address the impact of endothelial-specific ANGPTL4 ablation on vascular permeability in addition to 70kDa FITC-dextran and Ter119 (erythrocyte-specific) extravasation, we performed an Evans blue (EB) assay. We used intravenous injections of EB together with 4kDa FITC-dextran in WT and *Angptl4*^{ΔEC} mice bearing LLC isografts. Spectrophotometric measurements of extravasated EB showed significantly

lower levels of EB infiltrating the tumor in *Angptl4^{ΔEC}* mice compared to WT mice (**Supplementary Fig. 3f**). Similarly, fluorescent imaging of the 4 kDa FITC-dextran revealed decreased diffusion in the tumors from *Angptl4^{ΔEC}* mice (**Supplementary Fig. 3g**). This is consistent with the results shown in the original version of the manuscript. These new data have been included and discussed in the revised manuscript.

4. Is it possible for Lipase/FAO axis to regulate angiogenesis through controlling FOXO1/cMYC pathway?

The question raised by the Reviewer concerning the possibility that the Lipase/FAO axis regulates angiogenesis through the FOXO1/cMYC pathway is indeed intriguing. As detailed in the original version of our manuscript, we observed that the FOXO pathway is among the most significantly upregulated pathways following *ANGPTL4* knockdown (KD). The upregulation of this pathway strongly correlates with a decreased glycolytic phenotype. The Potente laboratory has thoroughly characterized the involvement of the FOXO1/cMYC pathway in regulating endothelial cell (EC) metabolic function in the context of angiogenesis (PMID: 26735015), as well as its role in maintaining a quiescent phenotype. Moreover, another recent study from the same group demonstrated that FOXO1 maintains EC quiescence by increasing the level of (S)-2-hydroxyglutarate ((S)-2HG). This supports the idea that FOXO1 regulates EC function at both the transcriptional and metabolomic levels, aligning with our findings (PMID: 33795871).

Although we have not specifically investigated the downstream effects of FOXO1/cMyc in controlling EC metabolism and angiogenesis after *ANGPTL4* silencing, we posit that the FOXO1/cMyc pathway plays a key role in controlling the expression of vital glycolytic genes, a pattern we consistently observed upon *ANGPTL4* knockdown in ECs.

Regarding whether the Lipase/Fatty Acid Oxidation (FAO) axis directly regulates angiogenesis through the FOXO1/cMYC pathway, we do not possess definitive evidence. Our RNA sequencing data shows that lipase inhibition does not appear to reverse the expression of FOXO pathway genes. The direct relationship between the lipase/FAO axis and angiogenesis through the FOXO1/cMYC pathway is still an area where concrete evidence is lacking in our research. This effect may be linked to other LIPG-independent functions of *ANGPTL4*, an area we intend to investigate further in the future.

5. Since EC proliferation also requires carbon contribution from FAO, it is confusing that XEN445 mediated FAO inhibition didn't show any anti-angiogenic effects by itself. What do the authors think is the difference?

We agree with the Reviewer that EC proliferation requires FAO in addition to glycolysis (PMID23911327, PMID25830893). Although ECs are primarily glycolytic, they do utilize FAO to some extent to maintain physiological function. The point that we are trying to convey here, is that the excess of FAs, due to increased lipolysis in the absence of *ANGPTL4*, “forced” their utilization via increasing their oxidation and inhibition of glycolysis. Since EC preferentially use glucose as an energy source and for vessel sprouting (PMID23911327), we also observed decreased angiogenic responses even though FAO was enforced (see above our response to second major issue). By using XEN445, we aimed to restore the inhibition of EL (lost after *ANGPTL4* KD). In this condition, we observed decreased lipase activity. This was accompanied with a reduction of FAs, FAO, as well as a partial restoration of glucose utilization and angiogenic responses. However, in non-silencing control conditions, expressing *ANGPTL4* [the inhibitor of EL], with the pharmacological inhibition XEN445, did not produce any significant effect. Similarly, silencing EL in EC expressing *ANGPTL4* did not produce any significant effect on EC metabolism and functions. We have further discussed these results together with the new results obtained in response to other points #6 (see below).

6. Rescue of key phenotypes (e.g. proliferation) should be tested by siLIPG. Specifically, proliferation/angiogenesis should be tested under conditions in Ext. data 5h.

As per the Reviewers suggestions, we also analyzed cell proliferation when *LIPG* was silenced. Data are included as **Supplementary Fig. 8h**) and accordingly described and discussed within the manuscript.

7. Lines 41 (abstract) and 463, 478, 493 in text mention that altered fatty acid metabolism promotes quiescence. This statement is too strong and should be reworded. Ref 42 is referenced, but the statement is not a correct interpretation of the reference: there is no evidence that FAO promotes quiescence.

We apologize for the overstatement and misleading interpretation. As indicated, we have reworded the above indicated statement and corrected the interpretation related to the original reference.

Minor

8. In line 371, confusing interpretation that “via inhibiting lipase activity”, is it “via increasing...”?

We apologize for the mistake. This has been corrected in the revised manuscript.

9. In Extended Fig 7c, the representative picture doesn't show consistent outcome with that of quantification on the right.

We have replaced original images that better reflect the quantification. (Original Supplementary Fig. 7c, present **Supplementary Fig. 10a**)

10. In Fig7e, the representative fig doesn't show consistent outcome with that of quantification on the right.

We have replaced original images that better reflect the quantification.

11. In line 540, the statement “promoting angiogenesis and tumour growth” is opposite to the data Fig2 shown in the manuscript.

We are grateful to the attention given to the details by the Reviewer; the statement has been corrected in the revised manuscript.

12. In line 156, Fig. 2a should be cited instead of Fig. 1a

We apologize for the mistake. The figure is now correctly cited.

13. Line 175 contains typo: ANTPL4 should be ANGPTL4

The typo has been corrected.

Reviewer #3:

The paper by Chaube and colleagues is the first to examine the metabolic and angiogenic effects of ANGPTL4 deficiency in endothelial cells (EC). The paper shows that silencing of ANGPTL4 in EC has major effects on a large number of metabolic and angiogenic parameters. The data point toward a huge role of ANGPTL4 in EC metabolism and angiogenesis, which is suggested to be mediated by the inhibitory action of ANGPTL4 on endothelial lipase (EL). Specifically, it is suggested that the inactivation of ANGPTL4 promotes EL-mediated vascular lipolysis, which results in increased fatty acid uptake and utilization in favor of FA oxidation. Although the link between ANGPTL4 and angiogenesis has been extensively studied, the role of ANGPTL4 in EC remains rather unclear, mostly because the published data are very conflicting. Several papers, including the first papers reporting on ANGPTL4 expression in endothelial cells, have found that ANGPTL4 stimulates in vitro tube formation and endothelial migration, as well as in vivo angiogenesis. By contrast, several other papers report an anti-angiogenic role for ANGPTL4. The inhibitory effect of ANGPTL4 on endothelial lipase has also been previously reported yet the impact of this pathway on EC metabolism and angiogenesis has not been studied. All previous studies that have investigated the role of ANGPTL4 in EC have used recombinant ANGPTL4 protein or an overexpression model. In contrast, Chaube and colleagues have inactivated the ANGPTL4 gene either via in vitro siRNA-based silencing or via in vivo Cre—mediated gene inactivation.

The main novelty of the paper is the finding that ANGPTL4 silencing appears to have a huge effect on glycolytic and fatty acid oxidative pathways, which is associated with major changes in the expression of numerous metabolic genes. Furthermore, it is shown that endothelial ANGPTL4 deficiency in vivo has anti-tumor and anti-angiogenic effects, which is explained by increased lipase activity, suggesting that local lipase activity in EC is important for the regulation of endothelial functions and tumor angiogenesis.

The paper has clear merit, as indicated above. My main concerns relate to the mechanistic framework for the paper. To my understanding, there is no precedent and no biochemical basis for EL regulating metabolic pathways in EC in the absence of external phospholipids or triglycerides as substrate. For example, Nakajima found that EL upregulates the expression of fatty acid oxidation-related enzymes and intracellular adenosine triphosphate accumulation, but these studies were done in the presence of high-density lipoprotein, which the main EL substrate (PMID: 23460280). Most of the experiments shown were conducted in EC in the absence of an EL substrate. The data by Chaube imply that the function of EL goes beyond being a lipase, for instance by showing that ANGPTL4 regulates the uptake of non-esterified fatty acids by EC via EL, a process that doesn't require a lipase enzyme. It is not impossible that the function of EL goes beyond that of a lipase but this needs to be shown in the paper to validate the suggested sequence of events occurring upon ANGPTL4 silencing.

We are grateful to the Reviewer for acknowledging that our study is the “first to examine the metabolic and angiogenic effects of ANGPTL4 deficiency in endothelial cells.” We were glad to read that the Reviewer also noted the novelty and merit of our present study while noting limitations and discrepancies with previous published work. However, the Reviewer also expressed some degree of confusion in the summary of the description of our findings that was later further articulated in the Reviewer’s major comments (see our initial response below and with more detail in our response to Reviewer’s specific comments). We are grateful for all of your insightful comments and recommendations. We believe that in responding to all of them, we have substantially improved our manuscript. Reviewer’s comments are italicized and followed by our answers.

Briefly, the Reviewer questioned the regulation of EL by ANGPTL4 in our settings and interprets that our data imply that “*the function of EL goes beyond being a lipase, for instance by showing that ANGPTL4 regulates the uptake of non-esterified fatty acids by EC via EL, a process that doesn't require a lipase enzyme.*” We consider that this thought is based on the consideration that our experiments were performed in the absence of external substrate. However, our experiments were/are performed in the presence of lipoproteins and therefore the availability of the substrate is guaranteed. We provide FPLC analysis of the

FBS that we use in our experiments (see **Rev #3 Fig. 1**). Therefore, our data do not go beyond what have been already published in relation to EL-mediated inhibition by ANGPTL4 (see below more comprehensive response to major comments #3 and #4). We provided a more detailed description of how the experiments were performed and we have carefully revised the manuscript to avoid any confusion describing the results specifying that ECs were cultured in the presence of media containing lipoproteins.

Major comments

1) What is the evidence that the observed changes in fatty acid and glucose metabolism upon ANGPTL4 silencing in EC account for the effects of ANGPTL4 silencing on angiogenic functions? Couldn't it be the other way around? Currently, the temporal relationship between the changes in EC function and metabolism is unclear. Based on the data presented, it cannot be concluded that metabolic rewiring precedes angiogenic changes, such as EC migration, tube formation etc. The conclusions of the paper should be redacted accordingly.

We fully agree with the Reviewer's comment, and we apologize for this misleading conclusion. This was also pointed out by Reviewer 1 (see our Response to primary issue of Reviewer 1). As we discussed, this could be a "chicken-and-egg" situation difficult to solve based on our present data. As indicated, we have revised the conclusions raised in our manuscript.

2) Previously, the C-terminal portion of ANGPTL4 was shown to bind to specific integrins, which likely accounts for several of the non-metabolic functions of ANGPTL4. Concerning angiogenesis, it was shown that ANGPTL4 counteracts hypoxia-driven vascular permeability through integrin $\alpha\beta3$ binding, modulation of VEGFR2-Src kinase signaling, and endothelial junction stabilization (PMID: 27577973). Another paper found that the C-terminal portion of ANGPTL4 binds and activates integrin $\alpha5\beta1$ -mediated Rac1/PAK signaling, ultimately leading to endothelial disruption (PMID: 21841165). Do the authors have specific data that rule out the role of integrin signaling in mediating the observed effect of ANGPTL4 silencing on angiogenic functions?

Our present results indicate that ANGPTL4 exerts its pro-angiogenic actions, at least in part, via the so called "metabolic actions" related to the inhibition of lipase activity. As pointed out by the Reviewer, the literature describing the role of ANGPTL4 in regulating angiogenesis and vascular permeability is conflicting. The contribution of ANGPTL4 in angiogenesis is controversial as both anti and pro-angiogenic effects of ANGPTL4 were observed in different pathophysiological settings (PMC1851201, PMC4435481, PMC2241731, PMID15870027, PMID10473614, PMC2794957, PMID4583458, PMID27577973). Moreover, ANGPTL4 has been described to promote and limit vascular permeability in different pathophysiological conditions, through different mechanisms. While initial studies suggested that ANGPTL4 (full-length) inhibited tumor metastasis by decreasing vascular permeability, later studies reported that ANGPTL4 (C-terminal) promoted vascular permeability and tumor metastases through integrin $\alpha5\beta1$ mediated Rac1/PAK signaling to weaken cell-cell contacts. Conversely, it was also described that ANGPTL4 (C-terminal) limited vascular permeability via $\alpha\beta3$ binding (PMC1693729, PMC6819094, PMID21841165, PMID27577973). More recent work shows that ANGPTL4 (full-length or C-terminal) leads to the breakdown of EC-EC junctions (PMC6819094), while a previous report showed that ANGPTL4 (whole body KO approach) limits vascular permeability by increasing adherens junctions, tight junction integrity and promoting pericyte coverage (PMC3196087). Our present results show that tumor vessels in *Angptl4* ^{Δ EC} mice have increased pericyte coverage (**Fig. 2i and Supplementary Fig. 3i**), thus potentially reducing tumor vascular leakage. However, we found decreased mRNA and protein surface levels of $\alpha\beta3$ in the absence of ANGPTL4 in ECs (**Fig 1f and Supplementary Fig. 1g**), suggesting that different mechanisms than those described previously might be operating when ANGPTL4 is KD in ECs. Although our results suggest that inhibition of lipase activity is involved in the regulation of angiogenic functions of EC, our data also suggest that inhibiting EL is not enough to rescue the expression of some critical angiogenic gene signatures and functions. However, it still remains unclear whether ANGPTL4 also regulates angiogenic function in ECs by binding to a specific receptor to exhibit its downstream signaling or whether it works as an intracellular modulator of different signaling cascades in ECs; or, alternatively, whether these processes are regulated in conjunction with our newly

described EC-lipase-mediated metabolic effects. However, the lack of mouse-specific antibodies limits our understanding of how different secreted forms of ANGPTL4 may interact with endothelial ANGPTL4 in the regulation of angiogenic responses. This would help to reconcile our understanding of some previous work that evaluated the paracrine actions of different forms of ANGPTL4 in EC functions. In the revised version of the manuscript, we extended the discussion of previous work in conjunction with our present results and indicating the limitations and potential future work that is needed.

3) The experiments in figure 5 that show increased lipase activity and fatty acid uptake in EC treated with ANGPTL4 siRNA were done in the presence of exogenous sources of TG and fatty acids, respectively. No exogenous sources of TG and fatty acids were seemingly provided in the ANGPTL4 silencing experiments shown in the previous figures. Accordingly, it is difficult to envision how, in the absence of any EL substrate, increased lipase activity and fatty acid uptake can account for the observed metabolic and angiogenic effect upon ANGPTL4 silencing.

We apologize for the confusion we have generated. As advanced above and based on your comments, we realized that we fell short in describing the conditions of our experiments involving the silencing of *ANGPTL4* in human ECs. Thus, there was a reasonable misperception that our conditions were in the absence of substrate. Indeed, we provided a very brief description of our transfection protocol, basically referring to our previous published work. In the transfection protocol with cationic lipid transfection reagents to perform the silencing, cells are incubated in Opti-MEM (Reduced Serum Medium) for 6h, since serum negatively affects transfection efficiency. After this period, medium is replaced by normal culture medium for HUVECs that contains 20% FBS for the indicated times (from 24 to 72h, with most of the experiments performed 60h post-transfection and, based on very efficient silencing). Thus, HUVECs are most of the time incubated in media containing lipoproteins. We provide FPLC analysis of the FBS that we used in our experiments that show the presence of lipoproteins, particularly HDL>LDL>VLDL (see **Rev #3 Fig. 1**). We also realized that in some instances we wrote that cells “transfected for 60 h” when we meant that cells were kept in EC culture media (20% FBS) for 60h after the initial transfection (6h). We should have written “60h post-transfection”, since this could have led to the impression that cells were kept in reduced serum media and thus without lipoproteins. Previous work has shown that EL has both TAG and phospholipase activity (PMID12032167). As such, after *ANGPTL4* KD, EL activity and lipoprotein lipolysis are increased, since there is substrate in the complete media. To measure enzyme activity 60h post-transfection, the culture media (containing 20%FBS) was heparinized and then collected to measure TAG or phospholipase activity using the appropriate substrates as described in Methods of the manuscript. As shown in **Fig. 5e**, absence of *ANGPTL4* increased both TAG and phospholipase activity. So, our data do not go beyond what have already been published in relation to EL-mediated inhibition by *ANGPTL4*. Overall, the involvement of EL is highlighted when the experiments were performed in the presence of XEN445 (see **Figures 5e-j**), as well as when we KD EL (see our response to #5 and new **Supplementary Fig. 7e, f** and **Supplementary Fig. 8b, f-h**). For the uptake of lipids and free FAs, see below for a more comprehensive response to major concern #4. To avoid confusion for the reader, in the revised version, we provided a more detailed description of how the experiments were performed and we have carefully revised the result section of the manuscript.

4) EL is a lipase enzyme and is not known to be directly involved in the uptake of free fatty acids by cells. Hence, how can the increased uptake of fluorescently-labeled fatty acids upon ANGPTL4 silencing be attributed to EL? The experiments with XEN445 are very useful and convincing but also raise major questions. How can an EL inhibitor reduce the uptake of a fluorescently-labeled fatty acid? How can an EL inhibitor regulate the expression of numerous metabolic genes (Fig 6c)? There is no mechanistic basis for EL impacting the cellular uptake of on-esterified fatty acids. The paper would be much more persuasive if data could be provided showing that overexpression or silencing of LIPG (without modulating ANGPTL4) influences the key pathways affected by ANGPTL4 silencing in figures 1,2 and 4.

Thank you for your insightful comments. We acknowledge that our previous description of the experimental approach and results might not have adequately elucidated the link between *ANGPTL4* silencing and the uptake of non-esterified free fatty acids (see also our response to major concern #3). Our experiments did not support that an increase in EL/lipase activity, due to *ANGPTL4* KD, directly facilitates the

uptake of FFAs into ECs. As advanced above, *ANGPTL4* KD was performed in the presence of lipoproteins (20% FBS). For the uptake of lipids at the end of incubation time (60h post-transfection), media was replaced with media containing an emulsion of ³H-Chylomicrons, ³H-Triolein, or fluorescently labeled palmitate. For the latter, the addition of palmitate was aimed to assess whether the increased CD36 levels, observed after *ANGPTL4* KD, was further facilitating FFA uptake. This hypothesis was grounded on previous research showing that heightened expression of CD36 can result in excess FA uptake (PMC6159965). Therefore, the absence of *ANGPTL4* increased the hydrolysis of lipoproteins, since there was substrate availability in the media, that facilitated the initial lipid uptake (**Fig. 5b** and **f**). The increased CD36 levels (**Fig. 5c** and **5h**) might lead to an increase of FFA uptake (**Fig. 5g**). Nevertheless, we did not aim to establish that heightened FA uptake was a direct result of increased EL activity due to the absence of endothelial *ANGPTL4*, but reflecting processes that are occurring because of *ANGPTL4* KD. The participation of EL is underlined when the experiments were performed in the presence of XEN445 (see Fig. 5 e-j). To further investigate this, and as indicated by the Reviewer, we conducted additional experiments silencing EL expression in HUVECs. Our latest results indicate that when EL expression is suppressed in *ANGPTL4*-deficient ECs, lipid/FA uptake decreases (**Supplementary Fig. 7e**), correlating with a reduction in CD36 levels. We believe these results provide a more comprehensive understanding of the process and hope they adequately address the Reviewer's concerns. New results are included, and described accordingly within the manuscript.

[Response redacted]

5) A crucial experiment to do is to silence *LIPG* in EC and determine whether the key effects of *ANGPTL4* siRNA described in figures 1,2 and 4 (especially RNAseq analysis) can be abolished by EL silencing. This experiment can reveal whether EL mediates the metabolic effects of *ANGPTL4* in EC. The present manuscript only shows the effects of silencing of *LIPG* on lipid accumulation in EC treated with *ANGPTL4* siRNA (Supplementary Fig. 5h). It is important to comprehensively study the effects of co-silencing of *ANGPTL4* and *LIPG*.

As per the suggestion, we performed co-silencing of *ANGPTL4* and *LIPG* to analyze some of key experiments, particularly [³H]tri olein/lipid uptake, FAO, glycolysis, the expression of key genes involved in metabolic regulation and angiogenesis, as well as the effect on EC proliferation. Briefly, we found that *LIPG* KD does not produce any significant effect on any of these processes, when compared to control cells expressing *ANGPTL4*. However, while the increased lipid uptake and FAO, mediated by *ANGPTL4* KD, was abolished to control levels when *LIPG* was co-silenced, the reduced glycolysis and proliferation were only partially restored. The effect on the expression of key metabolic genes was restored after *LIPG* co-silencing except for the EC glucose transporter *SLC2A1* (GLUT1) which was only partially restored. These results are in line with the preliminary observations that we obtained by using XEN445 (**Supplementary Fig. 8f**). Co-silencing of *ANGPTL4* and *LIPG* mediated a similar outcome on metabolism and angiogenic functions (see new **Supplementary Fig. 7** and **Supplementary Fig. 8b, f-h**). Altogether, the angiogenic or metabolic effects

produced by the absence of ANGPTL4 are mediated “to a great extent” by its action on lipase activity; however, our results also suggest that other mechanisms not directly related to its action on EL might additionally participate in the regulation of EC angiogenic responses (see our response to your major comments # 1 and # 2 and our response to primary and second major issues or other point #2 of Reviewer 2). These data are now included and described in **Supplementary Fig. 7 and 8** and also discussed in the revised manuscript.

6) Another important experiment to do is to study the effect of exogenous ANGPTL4. Specifically, in a new experiment, EC treated with ANGPTL4 siRNA should be co-treated with exogenous ANGPTL4 (or conditional medium of ANGPTL4-overexpressing cells vs control medium) to see if the effects of ANGPTL4 silencing on metabolic pathways, cell migration/adhesion, and angiogenesis can be abolished by external ANGPTL4. This experiment could be conducted with N-terminal ANGPTL4 and C-terminal ANGPTL4 to determine which part of the ANGPTL4 protein mediates the observed effects.

This is a very important point raised by the Reviewer and a major issue in the ANGPTL4 endothelial cell biology field (see also our response to your major comment #2). As described above, the literature is conflicting regarding the role of ANGPTL4 in regulating EC angiogenic functions and vascular permeability, particularly in regard to which precise form is involved in the regulation of these processes. Most of the so called “non-metabolic” effects have been ascribed to the C-terminal portion but opposing results have been reported, as above described (**see our response to point #2**). Additionally, there are also examples where the metabolic portion of ANGPTL4 (N-terminal or full length), based on its described role on inhibiting LPL, has been described to regulate “non-metabolic” effects (PMC9013473). As we described in the manuscript, most of these studies primarily involved exposure of endogenous ANGPTL4-expressing ECs to exogenous ANGPTL4, or exploiting whole-body KO mice, thus neglecting a potential EC-specific autonomous/autocrine role of ANGPTL4. In the present work, we found that suppressing *ANGPTL4* expression in ECs reduces their angiogenic functions, which supports a pro-angiogenic role for endothelial ANGPTL4. It is now appreciated that upon angiogenic stimulation, ECs can switch almost instantaneously to an activated, highly proliferative, and migratory state in response to angiogenic factors and that this angiogenic switch is reflected/accompanied by an EC metabolic switch (PMID: 31484054). Indeed, metabolic pathways are key regulators of many EC functions including inflammation, angiogenesis and barrier function, among others. In line with this, we also showed that endothelial ANGPTL4 participates in the regulation of EC angiogenic and metabolic switches at least in part via its action on lipase activity. However, we cannot disregard that other mechanisms might be operative since lipase inhibition or KD did not fully rescue all of the phenotypes observed upon ANGPTL4 KD (see also our response to your major comments #2 and #5). We agree with the Reviewer that elucidating what form/s is/are implicated in the regulation of these very intricated processes (see also our response to your major comment #1) is very interesting but at the same time very challenging, as will be appreciated in the description of some of the experiments we performed in an attempt to address your comment:

Since, at least *in vitro*, we can attribute the metabolic/angiogenic effects to reduced EL inhibition in the absence of ANGPTL4, we first aimed to determine what forms of ANGPTL4 were involved in this process. We first checked EL lipase activity in media obtained from Ctrl NS conditions and when ANGPTL4 was silenced. We found that EL activity was dramatically reduced upon EL KD and increased upon ANGPTL4 KD [Response redacted] We then collected medium from ANGPTL4 KD ECs (as maximal activity) and added increasing concentrations of different forms of recombinant ANGPTL4 (full length, N-terminal, or C-terminal) to test their inhibitory capacity. [Response redacted] both the full-length and N-terminal forms were efficient in inhibiting lipase activity (IC50=1µg/ml, ~23 and 58nM, respectively) whereas the C-terminal form inhibited lipase activity only at higher concentrations. These results somehow agree with a previous report that showed that ANGPTL4 efficiently inhibited EL activity using a cell-based activity assay overexpressing EL or recombinant EL (PMC8417300). However, it was not clear what specific form of ANGPTL4 was used in these assays, and the description of the methods of their ANGPTL4 recombinant protein isolation suggested that flagged C-terminal ANGPTL4 was likely employed, which should be considered when interpreting their results. Nevertheless, based on our own activity results, we performed ANGPTL4 KD to perform rescue experiments and analyzed the effect on glycolysis. [Response redacted]

[Response redacted]

Although these experiments were aimed to define the contribution of the different forms of ANGPTL4, they may not be physiologically relevant since they do not represent what HUVECs are indeed secreting and thus are likely within a supraphysiological range. Moreover, due to limitations in the availability of specific antibodies, it is difficult to trace accurately and quantitatively which forms of ANGPTL4 are secreted by HUVECs. Thus, supplementing culture media with recombinant ANGPTL4 may not accurately capture the physiological context. Furthermore, another potential limitation in the broader endothelial ANGPTL4 research landscape is the lack of knowledge regarding whether ANGPTL4-mediated effects are mediated in a paracrine, autocrine, and/or cell-intrinsic manner. Our *in vivo* data suggest that endothelial cell-derived ANGPTL4 is critical for maintaining angiogenic functions, distinct from contributions made by surrounding cells, (see our response to main comment for Reviewer 1 and new tumor angiogenesis results shown in **Supplementary Fig. 4**). Consequently, as the reviewer initially suggested, we chose to conduct rescue experiments using endothelial cell-derived conditioned media. Briefly, to assess whether endothelial cell-derived ANGPTL4 could rescue cellular processes that were altered upon its knockdown, we first cultured HUVECs transfected with or without siANGPTL4 in a hypoxia chamber (1% O₂) for 48 hours to collect conditioned medium enriched (hypoxia induced in NS Ctrl HUVECs) or deprived (hypoxia in siANGPTL4 HUVECs) of endothelial ANGPTL4 (see **Supplementary Fig. 5g**). We did not use CoCl₂ to induce hypoxia, in this case, to avoid its presence in conditioned media and potential confounding results when added back to HUVECs. As shown in **Supplementary Fig. 5h** and **i**, our data indicate that when HUVECs with ANGPTL4 KD were grown in ANGPTL4-enriched conditioned medium, they exhibited increased glycolysis and cell proliferation when compared to ANGPTL4 KD HUVECs grown conditioned medium lacking ANGPTL4, suggesting that endothelial-derived ANGPTL4 is capable of rescuing the glycolytic and proliferative phenotypes observed upon ANGPTL4 knockdown.

[Response redacted]

7) Since this paper is the first to report on the phenotype of EC-specific ANGPTL4 deficient mice, it would be of interest to have information on basic metabolic parameters, including plasma TAG, cholesterol, non-esterified fatty acids, and glucose. Having plasma lipoprotein profiles would be even better.

We thank the reviewer for suggesting the metabolic characterization of EC-specific ANGPTL4 deficient mice. We analyzed total plasma TAG, cholesterol, and HDL levels in both males and females fed a chow diet since all of our *in vivo* animal experiments were done in mice subjected to chow diet feeding. Briefly, we did not observe any significant effect on these parameters. We also did not notice any effect on fasting glucose. In line with total lipid levels, we did not find any effect on the lipoprotein profile. Accordingly, we did not find any differences, between genotypes, in lipid uptake from triolein in white adipose tissue and liver: therefore, indicating that the contribution of endothelial *ANGPTL4* to circulating lipids is negligible when compared to adipose- or liver-derived *ANGPTL4* (PMC5926923, PMC8409581, PMC8560380, PMC8041937, PMC9270256). This suggests that the effect observed *in vivo* (**Supplementary Fig. 9a-d**) are locally occurring in the activated endothelial angiogenic compartment. However, the participation of endothelial *ANGPTL4* under high fat diet feeding warrants future investigation. We have only included the total lipid as **Supplementary Fig. 9e**) and we provide the lipoprotein profile (only performed in males since there were no difference between sexes) and glucose levels for your evaluation (see **Rev #3 Fig.3**)

Rev #3 Fig. 3: The loss of ANGPTL4 function in ECs does not affect serum lipid profile. (A) TAG (right panel) and cholesterol (left panel) content of FPLC-fractionated lipoproteins from pooled plasma of overnight-fasted 8-10 week-old WT and *Angptl4*^{ΔEC} male mice fed a Chow Diet. (B) Blood glucose level in overnight-fasted 8-10 week old WT and *Angptl4*^{ΔEC} mice fed a Chow Diet.

8) The link between EC-derived ANGPTL4 and metabolic pathways is entirely based on studies in HUVEC using siRNA. The manuscript would greatly benefit from a comparative analysis of the gene expression profiles (RNAseq) of primary EC isolated from WT and *Angptl4*^{ΔEC} mice, confirming the key metabolic outcomes of ANGPTL4 silencing observed in HUVEC cells, including the stimulatory effect of ANGPTL4 deficiency on fatty acid oxidation genes and inhibitory effect on glycolysis genes.

We completely agree with the point raised by the Reviewer. As suggested, we analyzed via qRT-PCR the mRNA levels of key metabolic genes in mouse lung endothelial cells (MLECs) isolated and immortalized from WT and *Angptl4*^{ΔEC} mice. We checked genes involved in glucose uptake and glycolysis including *Glut1*, *Hk2*, *Pfkfb3*, *Ldha* as well as genes involved in FAO such as *Cpt1a* and *Acs1l*. Interestingly, the changes in the expression of these genes in the absence of *Angptl4* were qualitatively similar to those observed in *ANGPTL4* KD HUVECs. These results have been included and described (see **Supplementary Fig. 5f**) within the revised version of the manuscript.

9) The paper builds on evidence that ANGPTL4 is an EL inhibitor. However, this evidence is weak. Plasma HDL levels are not altered by ANGPTL4 deletion in mice or the E40K variant in humans consistent with reduced EL inhibition (plasma HDL levels are increased in E40K carriers rather than decreased). Also, a recent genetic mimicry analysis does not support EL inhibition by ANGPTL4 in humans, although the data do support EL inhibition by ANGPTL3/ANGPTL8 (PMID: 36372100). The paper should put more effort into investigating the potential inhibition of EL by ANGPTL4, using the EC-specific ANGPTL4-deficient mouse model or other models.

This is again a very interesting point raised by the Reviewer. Recent work has shown that ANGPTL4 inhibits EL in a cell-based activity assay overexpressing EL (PMC8417300). Also, EL has been described to have both phospholipase and TAG activity although the latter is less prominent (PMID12032167, PMC8483387, PMC5858721), which is in line with our present results. However, as pointed out by the Reviewer, a recent report does not support EL inhibition in humans and E40K ANGPTL4 variant showed increased HDL levels rather than reduced as would be expected from the lack of inhibition of EL. To this regard, HDL clearance is decreased in *EL*^{-/-} mice (PMC151412), while ANGPTL4 deletion in mice does not affect HDL levels (PMC8666355). Furthermore, there is a significant association with a SNP in EL and HDL levels in humans. As described above in response to major comment #7, we did not observe any effect on total lipid levels, neither on HDL levels nor other lipoproteins, suggesting that endothelial ANGPTL4 does not play a relevant role in the metabolism of lipoproteins when mice were fed a chow diet. Given that endothelial

ANGPTL4 is an induced protein upon EC activation (e.g., angiogenesis) our results suggest that angiogenic induction of ANGPTL4 helps to maintain reduced lipase activity in a local and autocrine manner to prevent FA uptake and oxidation to facilitate proper glucose utilization for angiogenic responses (PMID: 31484054, PMID23911327). As mentioned above, both the participation of endothelial ANGPTL4 under high fat diet feeding, as well as the effect on white adipose tissue expansion, a process that relies on angiogenesis, warrant future investigation. In the revised version of the manuscript, we have extended the discussion to accommodate the Reviewer's comments.

Minor comments

1) The paper would benefit from experiments in HUVEC cells cultured under hypoxic conditions, as opposed to using CoCl₂ to induce chemical hypoxia.

We completely agree with the Reviewer's comment. A decrease in oxygen concentration is the optimal hypoxia model. However, the use of CoCl₂-induced chemical hypoxia is a reliable model that allows the study of hypoxia. Although with some limitations, the major advantage is that it stabilizes HIF-1 α /2 α under normoxic conditions and prevents the rapid degradation of HIF-1 α /2 α by reoxygenation for several hours after removing the medium with cobalt, thus, it allows the experimenter to open the culture plate/dish/flask many times, perform changes in media and treatments, while maintaining the stabilization of HIF (PMID: 30484873). For instance, we would not have been able to perform any of the Seahorse studies to test the effect of hypoxia. Although there are differences between both models, the induction of the expression of genes involved in angiogenesis and metabolic genes (e.g., glycolytic genes) is similar between them in several cell types including endothelial cells, as well as the induction of ANGPTL4, among others (PMC6593883, PMC7152765). This together with the culture advantages discussed above, led us to select this approach over the decrease of oxygen concentration. To perform the experiments with conditioned media (see our response to point #6) we incubated the cells in a hypoxia chamber to increase ANGPTL4 levels.

2) The manuscript would benefit from data showing that there is a decrease in ANGPTL4 protein in the Cd45- and CD31+ cells isolated from the lung of ANGPTL4i Δ EC mice.

Thank you for your valuable suggestions. We agree with the Reviewer that it would be very interesting to know the protein levels of ANGPTL4 in *Angptl4*^{i Δ EC} mice. Detection of ANGPTL4 protein was also raised by Reviewer 1 (further point #3). Unfortunately, there are not reliable and specific antibodies to detect mouse ANGPTL4. This is one of the major limitations of our study. We have accordingly discussed this limitation in the discussion section of the revised manuscript.

3) The following question may be better experimentally studied in a future publication but perhaps the authors can already speculate in the discussion. How do the authors envision that increased fatty acid uptake following enhanced EL activity leads to the massive changes in gene expression observed in EC treated with ANGPTL4 siRNA? What transcription factor could for example mediate the major effects on glycolytic gene expression?

Thank you for the insightful question. Our current data suggest that silencing ANGPTL4 does lead to altered gene expression. Intriguingly, we have also observed an upregulation of FOXO family genes, which are known to regulate c-Myc and its downstream metabolic genes (PMID26735015) and Hippo signaling pathways, particularly the downregulation of transcriptional co-activators YAP1/TAZ which are very well known to regulate metabolic processes and angiogenesis (PMID35726026). For further details on this aspect, please refer to our response to reviewer #2, where we discuss the potential roles of FOXO1 and the Hippo signaling pathways in modulating gene transcription. It is worth noting that existing literature suggests that increased fatty acid metabolism helps maintain endothelial quiescence and redox balance, often through the activation of FOXO pathways (PMID26735015, PMID30146488). Therefore, while we do not know if FA might directly influence the changes in gene expression we've observed, they could potentially exert their effects indirectly via these signaling pathways, which we intend to explore further. Alternatively, increased FFA in ECs, potentially generated by LP hydrolysis, can serve as peroxisome proliferator-activated receptor (PPAR α) ligands, and thus might be directly implicated in maintaining quiescent, non-activated states, as shown for

inflammatory activation (PMC151409). We hope this clarifies our current understanding and future directions for this research. As suggested, we have discussed these points in the revised manuscript.

4) *There is a problem with the resolution of the bottom heatmap of figure 3c. Same for figure 1e.*

We apologize for this and now we have replaced the above-mentioned heatmap with another one with better resolution. As the data submitted for initial submission was low in Pdf size, this dampened the resolution of some of the figures. We are now submitting the figures with high resolutions.

5) *In the methods, please indicate which ANGPTL4 exon(s) is amplified by the qPCR that issued to validate the effective knock-down of ANGPTL4 in the Cd45- and CD31+ cells isolated from the lung of Angptl4 Δ EC mice.*

As suggested, we have now indicated the ANGPTL4 exon(s) that were amplified (Exon 5) by the qRT-PCR to validate the effective knock-down of ANGPTL4 in the CD45- and CD31+ cells isolated from the lung of *Angptl4 Δ EC* mice.

6) *Line 242, 255: shouldn't it be anaerobic glycolysis?7) It is incorrect to say that EC-derived ANGPTL4 is essential for EC lipolysis and FA uptake (lines 305-307). Rather, EC-derived ANGPTL4 may inhibit EC lipolysis and fatty acid uptake. Please adjust.*

We are grateful for catching these mistakes. We have now modified the manuscript as suggested.

REVIEWERS' COMMENTS

Reviewer #1 (Remarks to the Author):

Chaube and colleagues have submitted a revised version of their manuscript "Suppression of Angiopoietin-like 4 reprograms endothelial cell metabolism and inhibits angiogenesis". This new version includes additional data and explanations, which address the majority of my comments and concerns. Overall, the work is improved and - in my view - acceptable.

Reviewer #2 (Remarks to the Author):

The authors have satisfactorily addressed all issues. the only important point to tweak is the following: the inhibition of glucose metabolism by lipase activity is convincing, but it is not clear that this is mediated by increased fatty acid oxidation, versus other effects of increased fat influx (there are many). The text should therefore be careful not to imply this conclusion. e.g. the abstract "Knockdown of ANGPTL4 in ECs promotes endothelial lipase (EL)-mediated lipoprotein lipolysis, which results in increased fatty acid (FA) uptake and utilization in favor of FA oxidation hampering proper glucose utilization for angiogenic activation of ECs and maintaining a metabolically quiescent phenotype" is a run-on sentence that could be misconstrued to be saying that FAO is responsible for inhibition of glucose use and angiogenic activation, which is not demonstrated (and likely not the case).

Reviewer #3 (Remarks to the Author):

The manuscript has been appropriately revised. I have no further comments.

Point by point response for original submission (NCOMMS-22-49560A).

Point by point response to all of the comments raised the Reviewers.
Reviewer's comments are bold and italicized and followed by our answers.

Reviewer #1:

Chaube and colleagues have submitted a revised version of their manuscript "Suppression of Angiopoietin-like 4 reprograms endothelial cell metabolism and inhibits angiogenesis". This new version includes additional data and explanations, which address the majority of my comments and concerns. Overall, the work is improved and - in my view - acceptable.

We sincerely appreciate the Reviewer's comments, especially the recognition that our manuscript has improved and now is acceptable.

Reviewer #2:

The authors have satisfactorily addressed all issues. the only important point to tweak is the following: the inhibition of glucose metabolism by lipase activity is convincing, but it is not clear that this is mediated by increased fatty acid oxidation, versus other effects of increased fat influx (there are many). The text should therefore be careful not to imply this conclusion. e.g. the abstract "Knockdown of ANGPTL4 in ECs promotes endothelial lipase (EL)-mediated lipoprotein lipolysis, which results in increased fatty acid (FA) uptake and utilization in favor of FA oxidation hampering proper glucose utilization for angiogenic activation of ECs and maintaining a metabolically quiescent phenotype" is a run-on sentence that could be misconstrued to be saying that FAO is responsible for inhibition of glucose use and angiogenic activation, which is not demonstrated (and likely not the case).

We are grateful to the Reviewer for acknowledging that we have satisfactorily addressed all the issues. As indicated, we have revised the paragraph of the abstract to avoid any misleading conclusion. The revised paragraph is as follows:

"Knockdown of ANGPTL4 in ECs promotes lipase-mediated lipoprotein lipolysis, which results in increased fatty acid (FA) uptake and oxidation. This is also paralleled by a decrease in proper glucose utilization for angiogenic activation of ECs. Mice with endothelial-specific deletion of *Angptl4* showed decreased pathological neovascularization with stable vessel structures characterized by increased pericyte coverage and reduced permeability. Together, our study denotes the role of endothelial-ANGPTL4 in regulating cellular metabolism and angiogenic functions of EC."

We describe the observed effects, but we do not imply FAO is responsible for inhibition of glucose utilization and angiogenic activation.

We hope this is now acceptable.

Reviewer #3:

The manuscript has been appropriately revised. I have no further comments.

We are grateful to the Reviewer for considering that we have appropriately revised our manuscript.